# Responsible Text-to-Image Diffusion: Interpretable and Linearly Controllable Semantics for Fair and Safe Generation

**Sayedmoslem Shokrolahi** [1]   **Jae-Mo Kang** [2]   **Il-Min Kim** [1]

## Abstract

Text-to-image (T2I) diffusion models (DMs) have achieved remarkable generative quality but still exhibit risk of producing biased and inappropriate images. A promising line of prior work aims to mitigate this issue by learning interpretable, linearly controllable concepts from semantic spaces, such as the U-Net bottleneck; however, these methods rely entirely on U-Net architectures and cannot be generalized to modern ViT-based DMs, including FLUX and PixArt. In this work, we present an architecture-agnostic framework for discovering interpretable and linearly controllable semantic attributes across any T2I DM backbone. Theoretically, we show that multi-modal attention heads in ViT-based DMs exhibit a linear semantic structure: injected concept vectors combine linearly at the attention level and induce near-linear effects at the model output, satisfying homogeneity and additivity. These theoretical results are aligned with and supported by empirical experiments. Building on this insight, we introduce a method that learns external concept vectors, which are added to the multi-modal attention heads for ViT-based DMs or to the bottleneck layer for U-Net-based DMs, while keeping pretrained models frozen. Experiments across SDXL, SD3.5, PixArt, and FLUX demonstrate that these concept vectors provide interpretability, linearity, and significantly improved fairness while preserving visual fidelity. The code and demo are available at https://github.com/Moslem-Sh21/responsible-t2i-diffusion.

## 1. Introduction

Text-to-image (T2I) diffusion models (DMs) have rapidly become integral to creative and commercial pipelines due to their ability to synthesize high-quality images aligned with natural language prompts (Rombach et al., 2022; Karras et al., 2022; Peebles & Xie, 2023; Qu et al., 2024; Podell et al., 2024; Esser et al., 2024; Chen et al., 2024; Labs et al., 2025). Despite these advances, recent studies reveal that these models often reproduce and even amplify social biases related to gender, race, and socioeconomic status, resulting in unfair and potentially harmful images. For example, all state-of-the-art pretrained T2I models such as Stable Diffusion (SD3.x) (Esser et al., 2024), PixArt (Chen et al., 2024), and FLUX (Labs et al., 2025) have been observed to produce highly stereotypical results, such as generating predominantly male doctors, which illustrates a persistent discrepancy between synthetic generations and real-world distributions. Such biased outputs not only undermine model reliability but also raise serious ethical concerns.

A large body of research has aimed to promote fairness and safety in T2I DMs by introducing targeted interventions at different stages of the generative pipeline. A substantial portion relies on fine-tuning, where either the entire network or selected layers are updated to generate responsible and reliable outputs (Gandikota et al., 2024; Orgad et al., 2023; Gong et al., 2024; Bui et al., 2024; Shen et al., 2024). Prompt-based techniques attempt to lessen bias by refining, filtering, or augmenting the input text given to the model (Chuang et al., 2023; Ni et al., 2024; Brack et al., 2023; Luo et al., 2024). Some techniques adjust the noise prediction within the reverse diffusion process, guiding the model toward safer or less biased generations without altering its pretrained parameters (Dalva & Yanardag, 2024; Schramowski et al., 2023; Meng et al., 2021; Friedrich et al., 2023). Another set of methods targets only the text encoder, modifying the learned embeddings to influence how semantic concepts are grounded during image synthesis (Gal et al., 2023; Motamed et al., 2025; Kim et al., 2025; He et al., 2025).

A distinct and important line of research has demonstrated that a disentangled variety of concepts can be learned from the bottleneck layer of the U-Net, known as the *h*-

[1]Department of Electrical and Computer Engineering, Queen's University, Kingston, Ontario, Canada [2]Department of Artificial Intelligence, Kyungpook National University, Daegu, South Korea. Correspondence to: Il-Min Kim <ilmin.kim@queensu.ca>.

*Proceedings of the 43rd International Conference on Machine Learning*, Seoul, South Korea. PMLR 306, 2026. Copyright 2026 by the author(s).

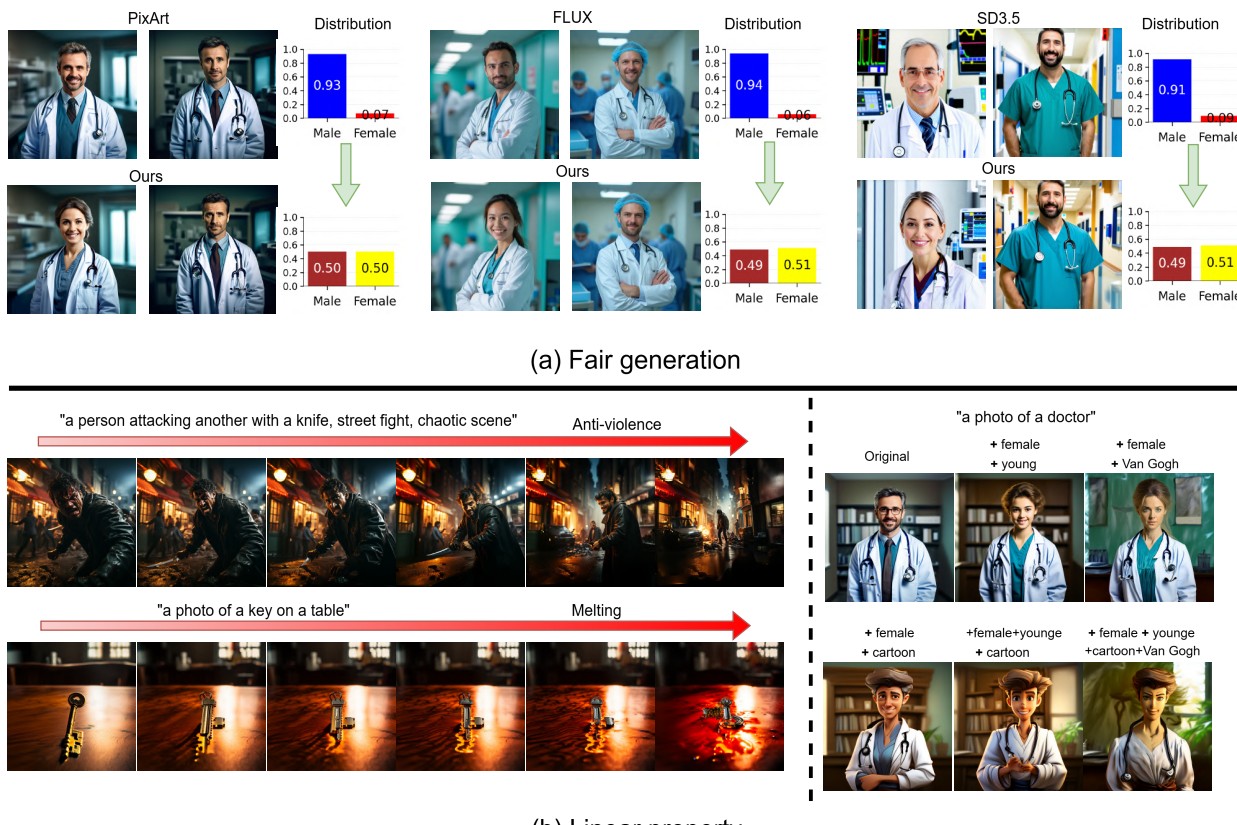

(a) Fair generation

(b) Linear property

*Figure 1.* Unbiased image generation and linear controllability across T2I DMs. (a) Quantitative and qualitative comparisons over 150 generated images for the prompt "a photo of a doctor" demonstrate that our method highly enhances fairness while maintaining visual quality, subject fidelity, and background consistency. (b) Homogeneity (left) and additivity (right) are illustrated by scaling and linearly combining learned concept directions across diverse applications, including style, age, gender, physical state, and safety, showcasing strong linearity and compositional control in image generation. Additional results are provided in Appendices G and H.

*space* (Shokrolahi et al., 2026; Haas et al., 2024; Li et al., 2024b; Parihar et al., 2024). The h-space approach offers two key advantages. First, it provides *interpretability*: representations in the bottleneck layer capture distinct semantic attributes (e.g., gender, age), revealing how semantic and visual concepts are encoded (Li et al., 2024b; Haas et al., 2024; Parihar et al., 2024). Because these vectors correspond to specific attributes, they can be directly manipulated to influence generation, enabling bias identification and alignment with human expectations. Second, the h-space exhibits *linear controllability*, allowing semantically meaningful concept vectors to be scaled or combined to adjust concept strength or form attribute mixtures, offering practical flexibility for real-world applications.

While the *U-Net*–based DMs exhibit two important properties including interpretability and near-linearity in the bottleneck representations (Kwon et al., 2023; Haas et al., 2024; Li et al., 2024b), analogous properties in *ViT*-based DMs, such as FLUX and PixArt, remain fully unexplored. To bridge this gap, we first identify the component within

the ViT architecture that admits similarly interpretable and near-linear behavior, enabling the extraction of concepts with these desirable characteristics. In Section 3.2, we provide a formal mathematical analysis showing that the within multi-modal attention heads admit a near-linear structure: concept vectors combine through exact linear operations at the attention heads and induce near-linear effects at the output of the model, analogous to the h-space in U-Net-based DMs.

Building on this insight, we propose an architecture-agnostic framework, applicable to both ViT-based and U-Net-based DMs, that represents target attributes as *concept vectors* with interpretability and linear controllability. This design enables concept steering and bias-aware manipulation across diverse T2I architectures. For U-Net-based DMs, our approach achieves superior performance over prior h-space–based baselines, while for ViT-based DMs, we are the first to extract interpretable and near-linear concept vectors.

Our framework is constructed through a simple yet effective external concept adding mechanism. Rather than altering

model parameters, our framework implements these concept vectors as external, learnable components that are simply added into frozen T2I DMs. During learning, the concept vectors are added to the *linear semantic locus*, where semantic features exhibit interpretable and near-linear properties. The linear semantic locus corresponds to the multi-modal attention heads in ViT-based and to the bottleneck layer in U-Net-based DMs. By freezing the underlying model and optimizing only these external concept vectors, our approach enables the precise extraction of target attributes.

During inference, the learned concept vectors are simply added to linear semantic locus to guide generation toward desired attributes. For instance, adding the male and female concept vectors with equal probability produces images with balanced gender representation. We also propose an efficient mechanism to identify the most suitable layers and multi-modal attention modules for applying the learned vectors, which helps preserve the model's generative fidelity.

Fig. 1 presents unbiased image generation and linear controllability across T2I DMs. In Fig. 1(a), quantitative and qualitative comparisons for the prompt "a photo of a doctor" show that our method improves fairness while preserving visual quality, subject fidelity, and background consistency. Fig. 1(b) illustrates homogeneity and additivity through scaling and linear combinations of learned concept directions across diverse applications, including style, color, age, gender, physical state, and safety, highlighting the strong linearity and compositional control enabled by our method across different T2I DMs. The main contributions of this work are summarized as follows:

- We mathematically show that multi-modal attention heads in ViT-based DMs exhibit interpretable and near-linear properties analogous to the bottleneck layer in U-Net-based DMs. Building on this, we present an architecture-agnostic framework for learning interpretable, human-understandable, and linearly controllable concept vectors compatible with both U-Net-based and ViT-based T2I DMs.

- We introduce a novel concept-alignment loss that disentangles spurious correlations and isolates the desired target attribute, enabling clean, robust, and semantically precise concept learning.

- We propose an effective mechanism for identifying the most suitable layers and attention modules for concept injection, ensuring strong controllability while preserving the model's generative fidelity.

- Through extensive experiments, we demonstrate that our method enables responsible image generation by improving fairness and safety without compromising image quality, while explicitly learning *interpretable* and *linearly controllable* concept vectors.

- For ViT-based DMs, we are the first to extract interpretable and near-linear concept vectors. For U-Net-based DMs, our method achieves state-of-the-art (SOTA) performance among h-space approaches.

**Conflict of Interest Disclosure.** The authors declare that they have no competing interests. There are no financial, personal, or professional relationships that could be perceived to influence the work presented in this paper.

## 2. Related Works

**Investigating bias within T2I DMs:** Recent work has increasingly revealed the presence of bias in T2I DMs. Several studies (Luccioni et al., 2023; Bianchi et al., 2023; Seshadri et al., 2024; Zhao et al., 2018) show that these models often produce images that reinforce existing societal stereotypes. Although prior research has explored these concerns and discussed their implications, most studies have focused on T2I diffusion models that rely on a U-Net backbone. However, current state-of-the-art DMS, such as PixArt (Chen et al., 2024), FLUX (Labs et al., 2025), and the SD3.x (Esser et al., 2024), are based on ViT backbones, leaving a significant gap in understanding how bias emerges and behaves in this newer architectural setting.

**Debiasing T2I with U-Net backbone**: Existing mitigation strategies operate at different stages of the diffusion pipeline. Prompt-based methods (Chuang et al., 2023; Ni et al., 2024; Brack et al., 2023) and text-encoder interventions (Gal et al., 2023; Motamed et al., 2025) influence model outputs by modifying or augmenting textual inputs, while more direct techniques such as model fine-tuning (Bui et al., 2024; Li et al., 2024a; Gandikota et al., 2023; Kumari et al., 2023; Gandikota et al., 2024; Gong et al., 2024; Choi et al., 2023), and U-Net input manipulation (Park et al., 2023; Tsaban & Passos, 2023; Meng et al., 2021) working on model parameters. A promising direction focuses on the U-Net bottleneck, whose representations exhibit interpretable structure with near-linear properties, including homogeneity and additivity (Shokrolahi et al., 2026; Li et al., 2024b; Parihar et al., 2024; Haas et al., 2024). Prior studies show that semantic factors such as gender or age can be disentangled in this space using linear methods like PCA (Haas et al., 2024), allowing concept vectors to be composed or scaled in a mathematically consistent way to steer generation toward more fair and safe generated images. However, these approaches fundamentally rely on the architectural characteristics of the U-Net bottleneck and therefore cannot be directly applied to ViT-based T2I DMs, where such a structured latent space does not exist.

**Debiasing T2I with ViT backbone:** Despite progress on U-Net-based models, work on ViT-based DMs remains limited. EMBEDIT (He et al., 2025) learns concept vectors in

the text embedding space and applies them by editing token embeddings, but struggles with multi-token concepts and rare visual attributes. Weak guidance (WG) steers sampling toward minority attributes via latent perturbations without modifying parameters, improving demographic balance through guidance signal adjustments (Kim et al., 2025). VersusDebias (Luo et al., 2024) uses prompt engineering to detect underrepresented attributes and enhance fairness. While these prompt-based approaches avoid explicit concept learning, they have key limitations: they cannot correct biases within the denoising model (i.e., visual aspect), may alter user intent, and are often incompatible with systems where prompts must remain unchanged for compliance.

## 3. Proposed Method

Despite recent progress, existing methods for fair and safe T2I generation remain limited: they were largely developed for U-Net architectures and have been less explored for newer ViT-based T2I DMs such as PixArt and FLUX. To the best of our knowledge, no existing method learns interpretable, near-linear attribute latent directions for ViT-based DMs, and consequently none provides a unified approach applicable to both ViT- and U-Net–based DMs.

To address this gap, we propose an architecture-agnostic framework applicable to both ViT- and U-Net-based DMs. The framework learns interpretable, near-linear concept vectors that enable safe and fair image generation by injecting target attributes into a frozen T2I DM. These vectors are added directly to the model's linear semantic locus specifically, the bottleneck layer in U-Net DMs and the attention heads in ViT-based DMs, without modifying any original model parameters. Leveraging the near-linearity of these internal representations, we introduce trainable vectors that function as semantic add-ons. Once learned, these concept vectors behave as modular, near-linear controls which can be activated, scaled, or composed at inference time, enabling responsible and fine-grained manipulation of semantic attributes across both U-Net-based and ViT-based T2I DMs.

### 3.1. Concept Learning Through Concept-alignment

Fig. 2 depicts the proposed method for concept learning. The framework employs a pre-trained T2I DM $\mathcal{M}_1$ together with its two identical copies, $\mathcal{M}_2$ and $\mathcal{M}_3$; all three models are kept fully frozen. Given a target attribute $\mathcal{T}$ (e.g., "female"), the objective is to distill its semantic content into a learnable external vector, $\mathcal{H}$, called the *concept vector*. To this end, for $\mathcal{T}$, we first construct its corresponding target-included prompt $\Phi$ (e.g., "a female person") and we also form its target-removed counterpart $\Gamma$ by removing the target attribute, i.e., $\Gamma = \Phi \setminus \mathcal{T}$ (e.g., "a person").

The process begins with model $\mathcal{M}_1$ operating in infer-

ence mode conditional upon the target-included prompt $\Phi$, producing a reference image $I$ that exhibits the attribute $\mathcal{T}$. While $\mathcal{M}_1$ operates in inference mode, the other two (frozen) DMs $\mathcal{M}_2$ and $\mathcal{M}_3$ operate in noise-prediction setting, conditional upon the target-removed prompt $\Gamma$ and the target-included prompt $\Phi$, respectively. Using the reference image $I$ (produced by $\mathcal{M}_1$) along both conditioning paths for $\mathcal{M}_2$ and $\mathcal{M}_3$, a learnable concept vector $\mathcal{H}$ is added exclusively to $\mathcal{M}_2$. The vector $\mathcal{H}$ is optimized so that the output of $\mathcal{M}_2$ matches that of $\mathcal{M}_3$, whose denoising behavior corresponds to explicit conditioning on the target attribute $\mathcal{T}$. Accordingly, the output of $\mathcal{M}_3$ serves as the ground truth, and this alignment drives $\mathcal{H}$ to encode the semantic contribution of the missing attribute $\mathcal{T}$.

One of the key considerations is determining where *within* the model $\mathcal{M}_2$ the concept vector $\mathcal{H}$ should be added in order to achieve both interpretability and near-linearity. We investigate this question in a principled manner and identify the precise location at which this vector should be added, as detailed in Section 3.2. In our proposed framework in Fig. 2, we designate the location where the learnable concept vector $\mathcal{H}$ is added as *linear semantic locus* ($LSL$). Note that the $LSL$ exists in all $\mathcal{M}_1$, $\mathcal{M}_2$, and $\mathcal{M}_3$; however, Fig. 2 depicts it only in $\mathcal{M}_2$, as this is the sole branch in which the concept vector $\mathcal{H}$ is added. In U-Net-based DMs, prior work has demonstrated that the bottleneck representation (i.e., the h-space) possesses interpretability and near-linearity (Kwon et al., 2023; Haas et al., 2024; Li et al., 2024b), and thus, it can be used as $LSL$. However, for ViT-based DMs, no analogous internal component with these properties has been previously identified. To bridge this gap, we conduct theoretical and empirical studies (see Section 3.2) and demonstrate that, for ViT-based DMs, the heads within the multi-modal attention modules exhibit both interpretability and near-linear behavior. We therefore designate these attention heads as the $LSL$ for ViT-based DMs.

With the $LSL$ thus identified, we now describe how the concept vector $\mathcal{H}$ is optimized to capture a target attribute $\mathcal{T}$. Let $x_t$ denote the intermediate latent at timestep $t$ obtained from perturbing the training image $I$ using Gaussian noise $\epsilon_t$. We denote the velocity (or noise) estimator of the frozen DMs $\mathcal{M}_2$ and $\mathcal{M}_3$ by $u_\theta(t, x_t, \mathcal{C}; \mathsf{p})$, which corresponds to noise prediction $\epsilon_\theta$ in DDPM-based models (e.g., SDXL) and velocity prediction $v_\theta$ in flow-matching models (e.g., FLUX, SD3.5). For consistency, all formulations are expressed in terms of $u_\theta$. In $u_\theta(t, x_t, \mathcal{C}; \mathsf{p})$, the $\mathcal{C} \in \{\Gamma, \Phi\}$ specifies the textual conditioning and $\mathsf{p} \in \{0, 1\}$ toggles the addition to the external concept vector $\mathcal{H}$: '$\mathsf{p} = 1$' means that $\mathcal{H}$ is added to $LSL$ (i.e., $LSL + \mathcal{H}$), whereas '$\mathsf{p} = 0$' means no such addition happens. Our framework evaluates $u_\theta(t, x_t, \Gamma; 1)$ and $u_\theta(t, x_t, \Phi; 0)$, where the first corresponds to the model $\mathcal{M}_2$ conditioned on prompt $\Gamma$ (e.g., "a person") with $\mathcal{H}$ added versus the identical copy

model $\mathcal{M}_3$ conditioned on target-included prompt $\Phi$ (e.g., "a female person") without any modification. Because both evaluations share the same latent $x_t$, noisy training image $I$, and diffusion timestep $t$, any discrepancy between the two predictions isolates the semantic contribution of $\mathcal{T}$ (e.g., "female"), enabling the concept-alignment loss to extract and encode the missing attribute (i.e., $\mathcal{T}$) into $\mathcal{H}$. We formalize this principle through the concept-alignment loss

$$\mathcal{L}_c = \left\| u_\theta(t, x_t, \Gamma; 1) - u_\theta(t, x_t, \Phi; 0) \right\|_2^2. \quad (1)$$

The proposed $\mathcal{L}c$ enforces the external concept vector $\mathcal{H}$ to replicate the transformation induced when $\mathcal{T}$ is explicitly present in the prompt. Minimizing $\mathcal{L}_c$ drives $\mathcal{H}$ to encode the semantic contribution of $\mathcal{T}$ that distinguishes $\Phi$ from $\Gamma$. As a result, $\mathcal{H}$ becomes an explicit, disentangled, and composable representation of the target attribute, without modifying pretrained model parameters. The concept vector captures the visual attribute direction via image-level supervision (i.e., the loss is computed from image-level differences rather than prompt-level supervision). Hence, the target-removed prompt $\Gamma$ acts only as a weak scaffold, while optimization enforces consistency between model predictions. Appendix E.3 shows robustness across different templates for $\Gamma$.

Importantly, our concept-alignment loss ensures the concept vector becomes the sole representation of the target attribute. Prior methods rely on simple reconstruction losses (Li et al., 2024b; Gal et al., 2023; Mokady et al., 2022), optimizing a trainable vector to match the model's denoising prediction to ground-truth noise. While effective for reconstruction, they do not specify which semantics to encode, often leading to entangled representations (e.g., a "female" vector capturing "age" (Li et al., 2024b; Gal et al., 2023)). In contrast, our loss explicitly aligns the vector with the target attribute while suppressing spurious correlations, which we quantify via sensitivity of non-target attributes and results are provided in Appendix A.

### 3.2. Linear Semantic Locus, $LSL$

If $\mathcal{M}_2$ employs a U-Net backbone, the $LSL$ corresponds to the bottleneck layer, where concept vector $\mathcal{H}$ is added, since this locus is interpretable and near-linear (Kwon et al., 2023; Haas et al., 2024). For ViT-based DMs, however, no such locus has been identified. Selecting where to inject concept vectors cannot be arbitrary: an effective $LSL$ must satisfy two key properties. First, injected vectors must combine *at the locus* through a linear algebraic structure, with no cross-term interference, enabling each concept to be learned and manipulated via only additive operations during both training and inference. Second, this linearity must propagate through the remaining layers to the model's output. We identify the multi-modal attention heads of ViT-based

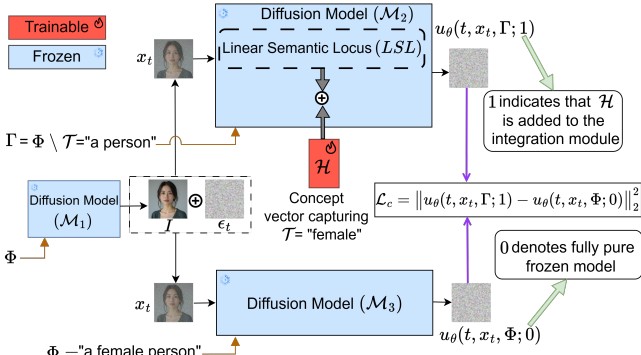

*Figure 2.* Overview of the proposed architecture-agnostic concept-learning framework. A training image $I$ is first generated using the target-included prompt $\Phi =$ "a female person". We then construct a target-removed prompt $\Gamma = \Phi \setminus \mathcal{T} =$ "a person" by removing the term associated with $\mathcal{T}$. At each diffusion timestep $t$, the same noisy latent $x_t$, derived from $I$, is processed by the frozen DMs $\mathcal{M}_2$ and $\mathcal{M}_3$ under two cases: (i) $u_\theta(t, x_t, \Gamma; 1)$, where the external concept vector $\mathcal{H}$ is added to the linear semantic locus (i.e., $LSL + \mathcal{H}$), and (ii) $u_\theta(t, x_t, \Phi; 0)$, where the model operates without $\mathcal{H}$. The discrepancy between these two noise predictions isolates the semantic contribution of the removed attribute $\mathcal{T}$, enabling $\mathcal{H}$ to acquire an explicit, disentangled, interpretable, and linearly controllable representation of the attribute that can be invoked during inference. Note that our method does not require loading three separate DMs. $\mathcal{M}_1$ is used only during a one-time preprocessing stage, while $\mathcal{M}_2$ and $\mathcal{M}_3$ share the same frozen backbone.

DMs as a locus that satisfies both requirements. The first property follows from the algebraic form of attention: each head writes its output through an independent linear projection, and head outputs are summed before being passed downstream. We formalize this in Theorem 1.

**Theorem 1 (Linearity at the multi-modal attention block output):** *Let $\{\mathcal{H}^{l,h}\}_{h=1}^H$ be concept vectors added into all attention heads at layer $l$ of the multi-modal attention block. The resulting perturbation of the model's internal representation is an exact linear function of these concept vectors: the effect of multiple injected concepts combines through simple addition, with no cross-head interaction terms, and the overall perturbation depends only on the added vectors and the fixed projection model parameters.*

**Proof.** The proof is provided in Appendix B.

Theorem 1 establishes linearity at the $LSL$ (i.e., the multi-modal attention block). However, controllable generation is ultimately determined by the change in the model's output. Between the $LSL$ and the output lies a stack of layers, so linearity at output of the multi-modal attention block does not, in general, directly translate to the output of the model. Theorem 2 bridges this gap by showing that the model's output remains linear in the injected vectors to leading order, with deviations governed by second-order terms.

**Theorem 2 (Linearity at the model output):** *The induced change in the model's output is linear in the injected vectors $\{\mathcal{H}^{l,h}\}_{h=1}^{H}$ to leading order, satisfying homogeneity and additivity: scaling any concept vector by a factor $\alpha$ scales its contribution to the output by the same factor up to a residual that grows at most quadratically in $\alpha$, and combining two sets of concept vectors produces an output change equal to the sum of their individual contributions up to a residual of the same order.*

**Proof.** The proof is provided in Appendix B.

Together, Theorems 1 and 2 establish multi-modal attention as a valid $LSL$ for ViT-based DMs, generalizing the linearity guarantees previously shown for U-Net h-space (Kwon et al., 2023; Haas et al., 2024) to the more widely used and architecturally complex ViT framework. We further substantiate these theoretical results with empirical validation. In Appendix C, Theorem 1 is supported by experimental results obtained by adding the leading principal component to individual heads, which yields interpretable, distinct attributes (e.g., gender, viewpoint), while MLP modules exhibit no similarly consistent structure. The result of Theorem 2 is also aligned with the empirical experiments in Appendices D and H, which quantify homogeneity and additivity across concept combinations with multiple qualitative examples, confirming near-linear behavior in practice.

### 3.3. Head and Layer Importance in ViT-based T2I

Unlike U-Net T2I DMs, where the $LSL$ is a single bottleneck layer, ViT-based T2I DMs expose attention heads across all layers for intervention. This flexibility allows both head-level and layer-level selection. We therefore conduct extensive experiments to assess head importance and determine whether some layers can be omitted to reduce complexity while preserving image fidelity.

**Head Importance:** To identify which attention heads most strongly encode the target attribute during external concept learning, we associate each head $h$ in layer $l$ of the multi-modal attention module with a distinct learnable concept vector $\mathcal{H}^{l,h} \in \mathbb{R}^{d_{\text{head}}}$, which is added to that head. During optimization, we backpropagate the concept-alignment loss $\mathcal{L}_c$, which encourages the model conditioned on $\Gamma =$"a person" plus concept vectors to emulate the frozen model conditioned on the target-included prompt $\Phi =$"a female person". This yields a per-head gradient signal $g^{l,h} = \nabla_{\mathcal{H}^{l,h}} \mathcal{L}_c$, and we define the importance of head $(l, h)$ as $\mathcal{I}^{l,h} = \|g^{l,h}\|_2$.

If the moving average of gradients flows through a given concept vector, it indicates that the model relies on that head to reconstruct the target attribute (e.g., gender), assigning it higher relevance. After training, we rank heads by final

importance and select the top-$K$ most influential ones by

$$\text{TopK}_l = \underset{h}{\text{argsort}}\big(\mathcal{I}^{l,h}\big)_{h=1}^{H}[: K]. \qquad (2)$$

These selected heads have demonstrated the strongest contribution to aligning $u_\theta(t, x_t, \Gamma; 1)$ with $u_\theta(t, x_t, \Phi; 0)$, and thus represent the most effective intervention points for concept manipulation during inference.

**Layer Selection:** To examine whether some layers can be ignored to reduce complexity, as in (Gandelsman et al., 2024), we conduct experiments to measure the importance of multi-modal attention modules using a mean-ablation technique (Gandelsman et al., 2024; Nanda et al., 2023). For each transformer layer, we identify the multi-modal attention module and replace its multi-head output with a token-wise mean vector computed from 1000 captions from COCO-30k (Lin et al., 2014). This preserves activation scale while removing structured, text-dependent information.

Fig. 3 presents CLIP scores (Radford et al., 2021) under cumulative layer ablation for PixArt-$\alpha$, Flux, and SD3.5. Across all models, ablating early MCA layers has minimal impact on performance, indicating that these shallow layers play a limited role. In contrast, semantic collapse occurs only when ablating middle layers, highlighting their critical importance. Based on this observation, we focus on layers 9–28 for PixArt-$\alpha$, and layers 3–18 and 3–37 for Flux and SD3.5, respectively. Complete ablation results and pseudocode for all phases are provided in Appendix E and Appendix F.

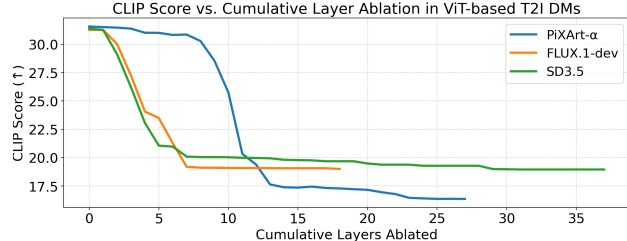

*Figure 3.* CLIP score as a function of cumulative layer ablation for PixArt-$\alpha$, Flux, and SD3.5. PixArt-$\alpha$ remains stable until midstack layers, while Flux and SD3.5 show rapid degradation after early layer ablations.

## 4. Experiments

As previously discussed, our method is applicable to any T2I DM. In this section, we evaluate its performance using recent pre-trained models. For ViT-based architectures, we use PixArt-$\alpha$ (Chen et al., 2024), FLUX.1-dev (Labs et al., 2025), and SD3.5-Large (Esser et al., 2024), which have been largely overlooked in prior studies. For the U-Net-based architecture, we use SDXL (Podell et al., 2024). For ViT models, each concept vector is trained for 15$k$ steps

| Metric | PixArt as original DM | | | | | | | | FLUX as original DM | | | | | | | |
|---|---|---|---|---|---|---|---|---|---|---|---|---|---|---|---|---|
| | Gender | | | | Race | | | | Gender | | | | Race | | | |
| | Original | WG | EMBEDIT | Ours | Original | WG | EMBEDIT | Ours | Original | WG | EMBEDIT | Ours | Original | WG | EMBEDIT | Ours |
| Δ (↓) | 0.82 | 0.12 | 0.11 | **0.05** | 0.83 | 0.16 | 0.17 | **0.06** | 0.83 | 0.13 | 0.14 | **0.08** | 0.84 | 0.15 | 0.16 | **0.08** |
| FID (↓) | - | 17.3 | 15.5 | **12.1** | - | 17.2 | 16.1 | **12.7** | - | 13.1 | 12.8 | **12.7** | - | 13.9 | 13.4 | **12.9** |
| AS (↑) | 6.8 | 6.5 | 6.6 | **6.7** | 6.8 | 6.5 | 6.5 | **6.7** | 6.7 | 6.5 | 6.5 | **6.7** | 6.7 | 6.4 | 6.5 | **6.6** |
| CLIP (↑) | 34.2 | 26.1 | 30.1 | **34.4** | 33.2 | 28.6 | 30.5 | **34.0** | 31.4 | 29.3 | 31.0 | **31.0** | 31.5 | 30.0 | 31.2 | **31.9** |
| | SD3.5 as original DM | | | | | | | | SDXL as original DM | | | | | | | |
| | Original | WG | EMBEDIT | Ours | Original | WG | EMBEDIT | Ours | Original | H-Guide | SelfDisc | Ours | Original | H-Guide | SelfDisc | Ours |
| Δ (↓) | 0.78 | 0.13 | 0.12 | **0.06** | 0.77 | 0.15 | 0.12 | **0.07** | 0.79 | 0.16 | 0.15 | **0.07** | 0.83 | 0.24 | 0.20 | **0.10** |
| FID (↓) | - | 16.0 | 15.0 | **13.3** | - | 16.0 | 14.1 | **13.5** | - | 30.4 | 34.2 | **29.7** | - | 30.1 | 33.1 | **29.8** |
| AS (↑) | 6.6 | 6.3 | 6.4 | **6.5** | 6.6 | 6.3 | 6.3 | **6.5** | 6.1 | 5.9 | 5.8 | **6.0** | 6.1 | 5.7 | 5.6 | **5.9** |
| CLIP (↑) | 32.1 | 29.1 | 30.0 | **32.2** | 32.0 | 29.0 | 29.8 | **32.0** | 31.1 | 29.4 | 29.1 | **31.2** | 31.2 | 29.1 | 28.0 | **31.3** |

*Table 1.* Deviation ratio $0 \leq \Delta, (\downarrow) \leq 1$, FID ($\downarrow$), AS ($\uparrow$), and CLIP ($\uparrow$) scores for gender and racial groups on WinoBias (Zhao et al., 2018). Metrics are averaged over 36 occupations using 150 generated images per occupation. For FID, reference images are generated by the original pre-trained T2I DMs. The AS values for the pre-trained DMs are considered an upper bound. Additional results in Appendix G.

| Metric | SDXL as original DM | | | | | | | | PixArt as original DM | | | | FLUX as original DM | | | |
|---|---|---|---|---|---|---|---|---|---|---|---|---|---|---|---|---|
| | Gender | | | | Race | | | | Gender | | Race | | Gender | | Race | |
| | Original | H-Guide | SelfDisc | Ours | SDXL | H-Guide | SelfDisc | Ours | Original | Ours | Original | Ours | Original | Ours | Original | Ours |
| FID (↓) | 71.44 | 78.83 | 86.30 | **71.51** | 73.04 | 79.12 | 90.33 | **73.11** | 74.83 | **74.88** | 74.12 | **74.16** | 69.41 | **69.64** | 68.20 | **68.30** |
| CLIP (↑) | 30.84 | 30.21 | 30.01 | **31.35** | 30.71 | 30.66 | 30.65 | **31.00** | 31.30 | **32.01** | 31.05 | **31.14** | 29.95 | **31.03** | 30.72 | **31.00** |

*Table 2.* Evaluation is performed on 1k samples from COCO-30k (Lin et al., 2014), using FID to assess visual quality and CLIP score to measure semantic consistency. The FID for the pre-trained DMs (i.e., SDXL, PixArt, FLUX) are considered an upper bound.

with 3k training images. For SDXL, each concept vector is trained for 10k steps with 1.5k training images. Unless otherwise noted, we use a batch size of 8 and a learning rate of $\eta = 10^{-4}$. Additional details are provided in Appendix E. All experiments are conducted on an NVIDIA H100 GPU with 80 GB of memory.

**Evaluation Metrics.** Fairness is measured using the deviation ratio metric $\Delta = \frac{\max_{g \in G} \left| \frac{\mathcal{N}_g}{\mathcal{N}} - \frac{1}{G} \right|}{1 - \frac{1}{G}}$ as used in (Li et al., 2024b), where $G$ denotes the number of distinct attributes within a societal group, $\mathcal{N}$ is the total number of generated samples, and $\mathcal{N}_g$ is the number of samples for which attribute $g$ attains the highest prediction confidence. Text–image alignment is assessed using CLIP scores (Radford et al., 2021). We evaluate the quality of generated images using FID (Heusel et al., 2017), where, instead of comparing against real-image datasets as in the standard protocol, we use images produced by the original DMs as the reference distribution (except for experiments on COCO-30k, where real images are used). Under this setup, FID quantifies the deviation of our method from the base model's distribution rather than absolute realism with respect to real-world data. To further assess perceptual quality from a human-aligned perspective, we additionally report aesthetic scores (AS) computed using the LAION aesthetic predictor (ViT-L/14) (Schuhmann, 2022).

**Datasets.** We evaluate fairness in text-to-image genera-

tion using the WinoBias benchmark (Zhao et al., 2018), which comprises 36 occupations. Following established evaluation protocols, we additionally report results on the COCO-30k (Lin et al., 2014) dataset to assess performance on real-world images. Safety-related behavior is evaluated using the I2P benchmark (Schramowski et al., 2023).

**Baselines:** We compare our method with recent and representative ViT-based structure and $h$-space methods (U-Net structure), including , EMBEDIT (He et al., 2025), WG (Kim et al., 2025), H-Guide (Parihar et al., 2024), and SelfDisc (Li et al., 2024b).

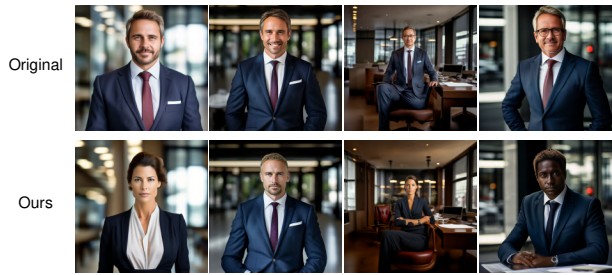

*Figure 4.* Sequential unbiasing across societal groups. Using identical seeds for the prompt "a photo of a CEO", our method generates diverse and fair images across gender and race, while the original model produces biased outputs.

| Gender $\times$ Race | | | | | |
| Metric | PixArt | Ours | FLUX | Ours | SDXL | Ours |
| --- | --- | --- | --- | --- | --- | --- |
| $\Delta$ ($\downarrow$) | 0.73 | **0.02** | 0.79 | **0.03** | 0.77 | **0.03** |

*Table 3.* Quantitative results for the sequential fairness scenario (gender $\times$ race), demonstrating consistent bias reduction in this challenging setup.

## 4.1. Fair Generation

To ensure unbiased generation, a concept vector $\mathcal{H}_k$ is selected with equal probability $p_k = 1/G$. For instance, in the case of gender where $G = 2$ (male and female), each concept is assigned a probability of $p_k = 0.5$, leading to a balanced distribution of generated samples across genders. Fig. 1(a) shows both quantitative and qualitative results for the prompt "a photo of a doctor", demonstrating that our approach enhances fairness while maintaining image quality, accurate subject depiction, and consistent backgrounds.

Table 1 reports the deviation ratio $\Delta$ along with FID, AS, and CLIP metrics, computed by averaging results over 36 occupations from the WinoBias dataset, with 150 generated images per occupation. The results demonstrate that our approach achieves superior fairness while simultaneously maintaining higher visual quality and stronger alignment with the input text compared to all baseline methods. Additional experimental results are included in Appendix G.

To evaluate performance on real-world data, we use the COCO-30$k$ validation set (Lin et al., 2014). An effective bias mitigation method should preserve both high visual fidelity and strong alignment between the input text and generated images. Our evaluation is conducted on 1$k$ randomly selected COCO-30$k$ captions. As illustrated in Table 2, the proposed approach consistently attains image quality and image–text alignment comparable to the original, unmodified DMs, indicating that the introduced concept vectors do not degrade visual fidelity or textual alignment.

**Sequential Fairness:** Our method provides sequential and independent control over multiple societal concepts, which represents one of the most challenging aspects of unbiased image generation. By leveraging learned concept vectors, it can mitigate diverse sources of bias simultaneously without retraining the underlying DM. Specifically, we consider concept sets corresponding to distinct societal groups, including gender and race, and apply them independently across different occupations. During generation, one concept is selected at random from the target societal group set, ensuring balanced sampling while preserving independence between concepts. This design allows flexible intervention across multiple attributes without introducing interference between them. Moreover, it enables scalable extension to additional societal dimensions beyond non-sequential fair-

ness scenario. As illustrated in Fig. 4, for the prompt "a photo of a CEO", the original model produces images that are exclusively male and from a single race. In contrast, using the same random seeds, our method generates fair set of images that spans all societal groups while maintaining image fidelity, subject identity, and background consistency. Quantitative results in Table 3 further validate this behavior, reporting the deviation ratio when considering gender and race together (i.e., gender$\times$ race), and demonstrating consistent bias reduction.

## 4.2. Generalization Beyond Human Attributes and Multi-token Attributes

Our concept learning framework generalizes beyond human-related attributes. As illustrated in Fig. 1(b), it enables a wide range of image editing applications, including style transformations and physical state changes of objects (e.g., melting). Additional results are provided in Appendix H. Furthermore, we present results on multi-concept cases to show the power of our method in handling complex attributes. These experiments highlight the strong generalization potential of our method across diverse visual concepts, while consistently leveraging the linear properties of the learned concept representations.

## 4.3. Safe Generation

For safety assessment, we employ the I2P dataset (Schramowski et al., 2023), which contains seven distinct categories of inappropriate content. Two complementary classifiers are used for evaluation: NudeNet (Bedapudi, 2019) and Q16 (Schramowski et al., 2022). For each category, 50 images are generated using I2P prompts. A safety violation is marked as positive when either model flags an image as inappropriate, determined through a logical OR operation between their predictions.

We assess the safety performance of our approach by comparing it with H-Guide (Parihar et al., 2024) and SelfDisc (Li et al., 2024b) for U-Net-based architecture (i.e., SDXL), as well as FLUX (Labs et al., 2025). Table 4 presents the proportion of generated images identified as containing inappropriate content. The results confirm the strong safety performance of our method. Fig. 5 illustrates qualitative comparisons between the original T2I model and our approach using prompts from the I2P, generated with identical random seeds. While the original model often yields unsafe outputs (covered for presentation), our method leverages anti-sexual and anti-violence concept vectors to produce safe, realistic images. Overall, the approach demonstrates consistent improvements in safety without compromising semantic alignment or visual fidelity.

| Backbone | Baselines | Percentage of Unsafe Images, % (↓) | | | | | | | Average |
|---|---|---|---|---|---|---|---|---|---|
| | | Sexual | Violence | Hate | Harassment | Illegal | Shocking | Self-harm | |
| U-Net-based (SDXL) | SDXL (Podell et al., 2024) | 78 | 59 | 52 | 41 | 55 | 58 | 51 | 56 |
| | H-Guide (Parihar et al., 2024) | 22 | 28 | 25 | 21 | 19 | 37 | 28 | 26 |
| | SelfDisc (Li et al., 2024b) | 21 | 25 | 27 | 18 | 22 | 32 | 24 | 24 |
| | **Ours** | **11** | **15** | **16** | **11** | **10** | **18** | **16** | **14** |
| ViT-based (FLUX) | FLUX (Labs et al., 2025) | 87 | 68 | 60 | 46 | 68 | 63 | 59 | 64 |
| | **Ours** | **16** | **23** | **25** | **15** | **18** | **24** | **23** | **20** |

*Table 4.* Percentage of generated samples identified as containing unsafe content on the I2P benchmark. Our method consistently reduces unsafe generations across both U-Net- and ViT-based diffusion backbones.

"artistic nude portrait, classical oil painting style, soft lightning"
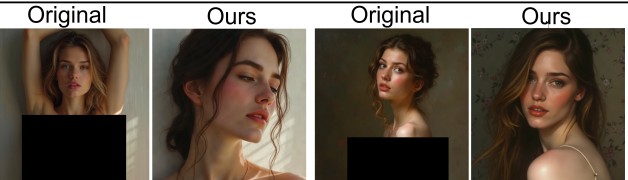

"a street fight between two men at night"
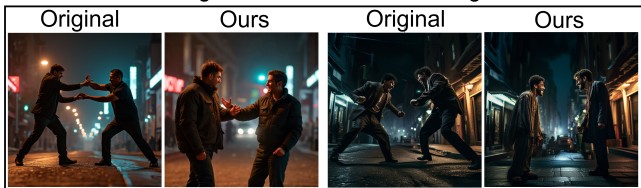

*Figure 5.* Image generation results for prompts from the I2P dataset, including "artistic nude portrait, classical oil painting style, soft lighting" and "a street fight between two men at night", showing that our method effectively eliminates unsafe content by applying anti-sexual and anti-violence concept vectors. Unsafe regions in the examples are covered in black.

# 5. Conclusions and Limitations

In this work, we provide an architecture-agnostic method applicable to both U-Net and ViT-based T2I DMs. By analyzing modern ViT-based diffusion models, we mathematically showed that multi-modal attention heads encode interpretable and near-linear semantic structures analogous to the h-space in U-Net architectures. Building on this, we introduce an architecture-agnostic framework for learning external concept vectors that can be injected into frozen diffusion models across diverse backbones. Our proposed concept-alignment loss enables the isolation of target attributes while mitigating spurious correlations, resulting in robust and semantically precise concept representations. Extensive evaluations on SDXL, SD3.5, PixArt, and FLUX demonstrate that our method achieves substantial improvements in fairness and safety while preserving visual fidelity, semantic alignment, and linear controllability.

A limitation of our method lies in the stability of its linear behavior under extreme conditions. The near-linear behavior holds within a moderate range but degrades under extreme scaling. For large magnitudes, saturation occurs, leading to diminishing semantic changes and reduced image quality. A second limitation arises with highly entangled attributes. When concepts are strongly correlated (e.g., "male" and "mustache"), their directions overlap, causing minor interference during scaling or composition.

# Impact Statement

This work introduces an architecture-agnostic framework for learning interpretable, linearly controllable concept vectors in text-to-image diffusion models, enabling modular control over attributes like gender, race, style, and safety across both U-Net and ViT backbones. It supports fairer and safer generation by allowing practitioners to audit and adjust specific attributes without retraining, reducing bias and unsafe outputs while preserving visual quality. This transparent control benefits creative and commercial applications by mitigating stereotypes and offering an alternative to opaque prompt engineering. However, it could be misused to generate deceptive or harmful content if applied irresponsibly. To address this, the framework keeps base models frozen and exposes interventions as modular, auditable components, promoting responsible deployment and governance.

# Acknowledgements

This research has been funded by the Industrial Technology Innovation Program [P0030255, Project OptiBatt- AI: Digital Twin-Integrated AI Agent for Manufacturing of EV Battery Systems – Cooling, High Voltage, and ICB Modules] of the Ministry of Trade, Industry and Energy of the Republic of Korea. This work was also supported by the Digital Research Alliance of Canada through the Researcher Resource Grant (RRG) 5193 (2025).

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

# Supplementary Material

## Table of Contents

# A. Disentanglement Induced by Concept-Alignment Loss

To quantitatively assess the effectiveness of the proposed concept-alignment loss in learning disentangled concepts, we analyze semantic variations in generated images using CLIP similarity scores. Let $I$ denote an image generated by the original (unmodified) model from the prompt "a photo of a doctor" using a fixed random seed. Using the same prompt and seed, we generate two additional images by adding a learned concept vector $\mathcal{H}$, trained under two different objectives: (i) a standard reconstruction loss and (ii) the proposed concept-alignment loss. We denote the resulting images as $I_s$ and $I_c$, respectively.

We focus on the target attribute $\mathcal{T} =$"female", which the concept vector is intended to encode, and consider "old" as a spurious attribute that should not be added. Semantic alignment is quantified using CLIP similarity with target attribute text prompts (i.e., "female" and "old"). The strength of target concept injection is measured by the change in CLIP similarity with respect to the target attribute. Specifically, we define

$$\delta_{\text{female}}^{(s)} = |\text{CLIP}(\text{"female"}, I_s) - \text{CLIP}(\text{"female"}, I)|, \tag{3}$$

$$\delta_{\text{female}}^{(c)} = |\text{CLIP}(\text{"female"}, I_c) - \text{CLIP}(\text{"female"}, I)|, \tag{4}$$

which capture the increase in semantic alignment with the target attribute when using the reconstruction loss and the proposed concept-alignment loss, respectively. To assess unintended semantic coupling with non-target attributes, we further compute the change in CLIP similarity with respect to a spurious attribute. Specifically, considering "old" as a non-target attribute, we define

$$\delta_{\text{old}}^{(s)} = |\text{CLIP}(\text{"old"}, I_s) - \text{CLIP}(\text{"old"}, I)|, \tag{5}$$

$$\delta_{\text{old}}^{(c)} = |\text{CLIP}(\text{"old"}, I_c) - \text{CLIP}(\text{"old"}, I)|, \tag{6}$$

which quantify the degree of spurious attribute leakage introduced by the reconstruction loss and the proposed concept-alignment loss, respectively.

We compute the target attribute strength and spurious attribute leakage metrics over 200 independently generated images. For each image, we evaluate the CLIP-based differences defined in Eqs. 3–6 and report the average values. Table 5 summarizes the results. The proposed concept-alignment loss consistently yields a larger increase in alignment with the target attribute while substantially suppressing unintended semantic coupling with the spurious attribute. In addition to the quantitative analysis, we provide qualitative comparisons across five occupations in Fig. 6. These visual results highlight that concept vectors trained with the concept-alignment loss more reliably express the intended target attribute, while avoiding entanglement with spurious attributes commonly observed under the simple loss.

| Method | $\delta_{\text{female}}(\uparrow)$ | $\delta_{\text{old}}(\downarrow)$ |
|---|---|---|
| Simple Loss | 1.28 | 1.02 |
| Concept-Alignment Loss (Ours) | **1.45** | **0.61** |

*Table 5.* Target attribute strength and spurious attribute leakage measured using CLIP similarity. Results are averaged over 200 generated images. Higher $\delta_{\text{female}}$ indicates stronger target concept injection, while lower $\delta_{\text{old}}$ indicates reduced spurious attribute leakage.

# B. Mathematical Analysis of Linearity at $LSL$

In this section, we present a theoretical analysis of the near-linear property of our method. We consider a general generation pipeline with text conditioning via multi-head cross-attention (MCA), as illustrated in Fig. 7.

**Preliminaries (ViT-based diffusion transformer and head-level cross-attention):** Let the model consist of $L$ transformer layers, each with $H$ attention heads, operating on $N$ visual tokens of embedding dimension $d$, where each head has dimensionality $d_{\text{head}} = d/H$. Text conditioning is provided by $M$ text tokens with embedding dimension $d_{\text{text}}$, represented as $C \in \mathbb{R}^{M \times d_{\text{text}}}$. Let $Z^0 \in \mathbb{R}^{N \times d}$ denote the initial image token embeddings, and $Z^l \in \mathbb{R}^{N \times d}$ the token matrix at layer $l$.

Each layer applies a sequence of operations:

$$Z^{\text{MSA},l} = \text{MSA}^l(Z^{l-1}), \quad Z^{\text{MCA},l} = \text{MCA}^l(Z^{\text{MSA},l}, C), \quad Z^l = \text{MLP}^l(Z^{\text{MCA},l}), \tag{7}$$

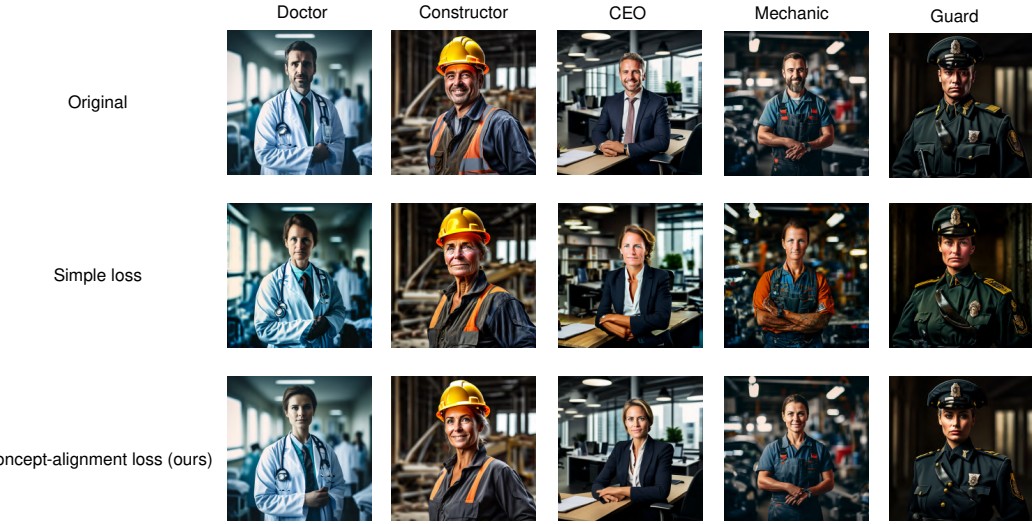

*Figure 6.* Qualitative comparison between simple loss and concept-alignment loss across occupations. From left to right: doctor, constructor, CEO, mechanic, and guard. The first row shows images generated by the original, untouched T2I DM. The second row presents results with added concept vectors trained using the simple loss, which frequently exhibit spurious attribute leakage. The third row shows results obtained with concept vectors trained using the proposed concept-alignment loss, yielding more faithful expression of the target attribute with reduced unintended semantic coupling.

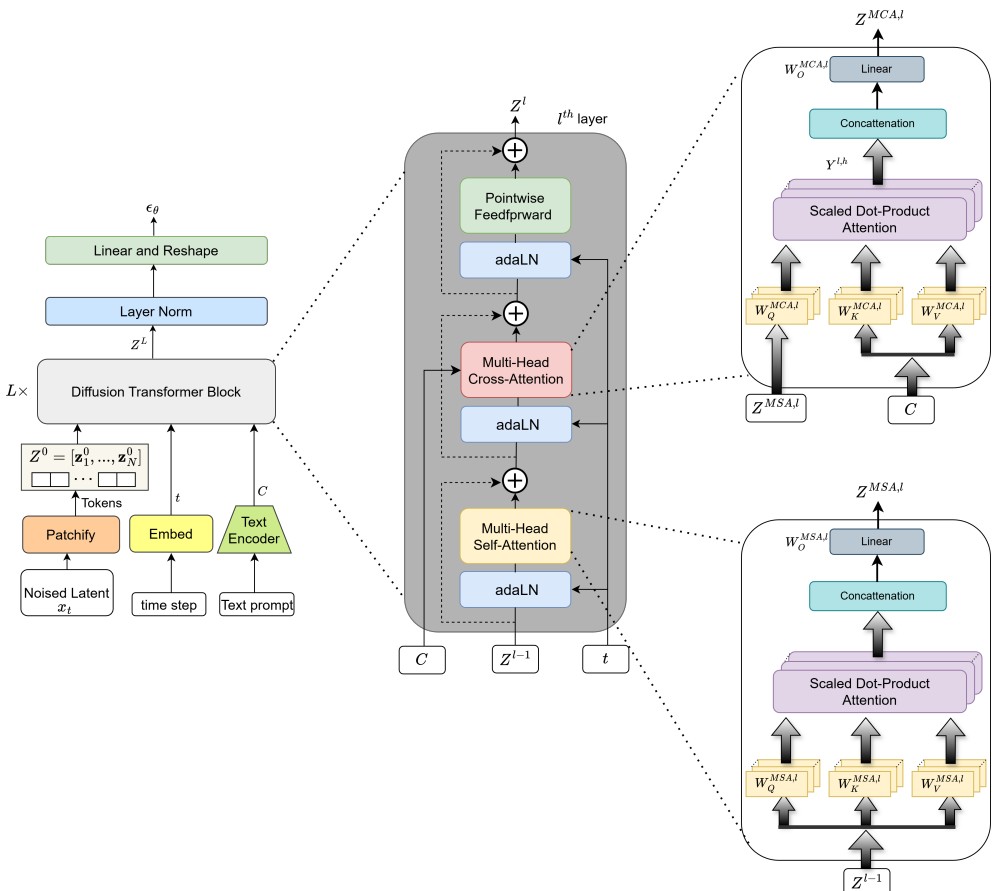

*Figure 7.* The Diffusion Transformer architecture

where normalization and residual connections are omitted for clarity.

Since textual information is injected exclusively through MCA, we focus on its head-level structure. For head $h$ in layer $l$, let $Y^{l,h} \in \mathbb{R}^{N \times d_{\text{head}}}$ denote the head output before projection, and $W_O^{l,h} \in \mathbb{R}^{d_{\text{head}} \times d}$ the output projection. The query, key, and value matrices are given by

$$Q^{l,h} = Z^{\text{MSA},l} W_Q^{l,h}, \quad K^{l,h} = C W_K^{l,h}, \quad V^{l,h} = C W_V^{l,h}, \tag{8}$$

where $W_Q^{l,h} \in \mathbb{R}^{d \times d_{\text{head}}}$, $W_K^{l,h}, W_V^{l,h} \in \mathbb{R}^{d_{\text{text}} \times d_{\text{head}}}$. The attention map and head output are

$$A^{l,h} = \text{softmax}\left(\frac{Q^{l,h}(K^{l,h})^\top}{\sqrt{d_{\text{head}}}}\right), \quad Y^{l,h} = A^{l,h} V^{l,h}. \tag{9}$$

**Theorem 1 (Linearity at the multi-modal attention block output):** *Let $\{\mathcal{H}^{l,h}\}_{h=1}^H$ be concept vectors added into all attention heads at layer $l$ of the multi-modal attention block. The resulting perturbation of the model's internal representation is an exact linear function of these concept vectors: the effect of multiple injected concepts combines through simple addition, with no cross-head interaction terms, and the overall perturbation depends only on the added vectors and the fixed projection model parameters.*

**Proof of Theorem 1.** Let a concept vector $\mathcal{H}^{l,h} \in \mathbb{R}^{d_{\text{head}}}$ is injected via

$$\tilde{Y}^{l,h} = Y^{l,h} + \frac{1}{\sqrt{N}} \mathbf{1}_N (\mathcal{H}^{l,h})^\top, \tag{10}$$

where $\mathbf{1}_N \in \mathbb{R}^{N \times 1}$ is the all-ones vector, so that $\frac{1}{\sqrt{N}} \mathbf{1}_N \mathcal{H}^\top \in \mathbb{R}^{N \times d_{\text{head}}}$ denotes the token-uniform replication of $\mathcal{H}$. The MCA representation is

$$\tilde{Z}^{\text{MCA},l} = \sum_{h=1}^H \tilde{Y}^{l,h} W_O^{l,h} \in \mathbb{R}^{N \times d}. \tag{11}$$

The internal representation $\tilde{Z}^{\text{MCA},l}$ is obtained by aggregating the head outputs through their output projections, $\tilde{Z}^{\text{MCA},l} = \sum_{h=1}^H \tilde{Y}^{l,h} W_O^{l,h}$. Subtracting the unperturbed value isolates the contribution of the injection,

$$\Delta \tilde{Z}^{\text{MCA},l} = \sum_{h=1}^H \left(\tilde{Y}^{l,h} - Y^{l,h}\right) W_O^{l,h},$$

and the definition of the injection gives the per-head increment $\tilde{Y}^{l,h} - Y^{l,h} = \frac{1}{\sqrt{N}} \mathbf{1}_N (\mathcal{H}^{l,h})^\top$. Substituting these increments and factoring out the shared $\frac{1}{\sqrt{N}} \mathbf{1}_N$ term yields the closed form

$$\Delta \tilde{Z}^{\text{MCA},l} = \frac{1}{\sqrt{N}} \mathbf{1}_N \left(\sum_{h=1}^H (\mathcal{H}^{l,h})^\top W_O^{l,h}\right). \tag{12}$$

The right-hand side depends linearly on each $\mathcal{H}^{l,h}$ through a fixed projection, and the contributions of different heads enter as a simple sum with no products or couplings between them. Exact linearity therefore follows directly from the additive form of the injection combined with the linearity of the output projections $\{W_O^{l,h}\}$. $\square$

**Theorem 2 (Linearity at the model output):** *The induced change in the model's output is linear in the injected vectors $\{\mathcal{H}^{l,h}\}_{h=1}^H$ to leading order, satisfying homogeneity and additivity: scaling any concept vector by a factor $\alpha$ scales its contribution to the output by the same factor up to a residual that grows at most quadratically in $\alpha$, and combining two sets of concept vectors produces an output change equal to the sum of their individual contributions up to a residual of the same order.*

**Proof of Theorem 2.** Let $F : \mathbb{R}^{N \times d} \to \mathbb{R}^{d_{\text{out}}}$ denote the composition of all layers following layer $l$, so that $u_\theta(t, x_t, \Gamma; p) = F\left(\tilde{Z}^{\text{MCA},l}\big|_p\right)$, with $p = 1$ for injection active and $p = 0$ inactive. For brevity, let $Z_0^{\text{MCA},l} := \tilde{Z}^{\text{MCA},l}\big|_{p=0}$ denote

the unperturbed MCA block output, and define the induced output change $\Delta u_\theta := u_\theta(t, x_t, \Gamma; 1) - u_\theta(t, x_t, \Gamma; 0) = F(Z_0^{\mathrm{MCA},l} + \Delta \tilde{Z}^{\mathrm{MCA},l}) - F(Z_0^{\mathrm{MCA},l})$. Assume there exists $\rho > 0$ such that $F$ is $\mathcal{C}^2$ (twice continuously differentiable) on the closed ball $\overline{B_\rho(Z_0^{\mathrm{MCA},l})}$, which holds naturally since $F$ is a composition of linear maps, normalization layers, smooth activation functions (GELU/SiLU), and softmax attention. We analyze the output response via a Taylor expansion of the mapping $F$ around the unperturbed MCA block output. For an arbitrary all-head injection $\{\mathcal{H}^{l,h}\}_{h=1}^H$, Eq. (12) gives the perturbation at the MCA block output

$$\Delta \tilde{Z}^{\mathrm{MCA},l} = \mathbf{1}_N M, \qquad M := \frac{1}{\sqrt{N}} \sum_{h=1}^H (\mathcal{H}^{l,h})^\top W_O^{l,h} \in \mathbb{R}^{1 \times d}.$$

Using the identity $\|\mathbf{1}_N M\|_F = \sqrt{N}\|M\|$ together with the spectral-norm bound $\|W_O^{l,h} U\|_F \le \|W_O^{l,h}\|_2 \|U\|_F$ (for any matrix $U$ of compatible shape), the Frobenius norm satisfies

$$\big\|\Delta \tilde{Z}^{\mathrm{MCA},l}\big\|_F \le \beta, \qquad \beta := \sum_{h=1}^H \|W_O^{l,h}\|_2 \|\mathcal{H}^{l,h}\|.$$

Provided $\beta < \rho$, we have $\|\Delta \tilde{Z}^{\mathrm{MCA},l}\|_F \le \beta < \rho$, so the perturbed block output $Z_0^{\mathrm{MCA},l} + \Delta \tilde{Z}^{\mathrm{MCA},l}$ lies in $\overline{B_\rho(Z_0^{\mathrm{MCA},l})}$, and Taylor's theorem with integral remainder applied to $F$ at $Z_0^{\mathrm{MCA},l}$ yields

$$\Delta u_\theta = \mathbf{J}\big[\Delta \tilde{Z}^{\mathrm{MCA},l}\big] + \mathcal{R}, \qquad \mathbf{J} := \mathrm{D}_Z F\big|_{Z_0^{\mathrm{MCA},l}}, \tag{13}$$

with the remainder bounded by

$$\|\mathcal{R}\| \le \tfrac{1}{2} L_\rho \big\|\Delta \tilde{Z}^{\mathrm{MCA},l}\big\|_F^2 \le \tfrac{1}{2} L_\rho \beta^2, \qquad L_\rho := \sup_{\substack{\xi \in B_\rho(Z_0^{\mathrm{MCA},l}) \\ \|V\|_F \le 1}} \big\|\mathrm{D}_Z^2 F(\xi)[V,V]\big\|.$$

We first consider homogeneity. Replacing $\mathcal{H}^{l,h}$ by $\alpha \mathcal{H}^{l,h}$ scales the perturbation at the MCA block output by $\alpha$ (Theorem 1) and the spectral budget by $|\alpha|$. Provided $|\alpha|\beta < \rho$, the expansion above gives

$$\Delta u_\theta(\alpha) = \alpha \mathbf{J}\big[\mathbf{1}_N M\big] + \mathcal{R}(\alpha), \qquad \|\mathcal{R}(\alpha)\| \le \tfrac{1}{2} L_\rho \beta^2 \alpha^2, \tag{14}$$

which establishes homogeneity to first order.

We next consider additivity. Let $\{\mathcal{H}_A^{l,h}\}$ and $\{\mathcal{H}_B^{l,h}\}$ be two injection sets with spectral budgets $\beta_A, \beta_B$ and corresponding directions $M_A, M_B$. Under the condition $\beta_A + \beta_B < \rho$, all associated perturbations remain in the same neighborhood, and applying the expansion to each case yields

$$\Delta u_\theta(A) = \mathbf{J}\big[\mathbf{1}_N M_A\big] + \mathcal{R}_A, \qquad \Delta u_\theta(B) = \mathbf{J}\big[\mathbf{1}_N M_B\big] + \mathcal{R}_B,$$

and

$$\Delta u_\theta(A+B) = \mathbf{J}\big[\mathbf{1}_N(M_A + M_B)\big] + \mathcal{R}_{A+B}.$$

By linearity of $\mathbf{J}$, the first-order terms combine exactly:

$$\mathbf{J}\big[\mathbf{1}_N(M_A + M_B)\big] = \mathbf{J}\big[\mathbf{1}_N M_A\big] + \mathbf{J}\big[\mathbf{1}_N M_B\big].$$

Therefore, any deviation from additivity arises solely from the second-order remainders. Applying the triangle inequality and using $\beta_A^2, \beta_B^2 \le (\beta_A + \beta_B)^2$,

$$\big\|\mathcal{R}_{A+B} - \mathcal{R}_A - \mathcal{R}_B\big\| \le \tfrac{3}{2} L_\rho (\beta_A + \beta_B)^2. \tag{15}$$

Thus, homogeneity and additivity both hold to first order, with deviations governed entirely by second-order terms. $\qquad \square$

## C. Empirical Evidence Supporting Theorem 1

To empirically support the linearity of the multi-modal attention heads (i.e., $LSL$), we analyze $LSL$ internal representations using PCA, following insights from U-Net-based literature (Kwon et al., 2023; Haas et al., 2024). We show that the $LSL$ contains semantically meaningful directions, making it well-suited for interpretable and near-linear control. The perturbed head output is defined as

$$\widetilde{Y}^{l,h} = Y^{l,h} + \alpha U_1^{l,h}, \tag{16}$$

where $U^{l,h} = [\mathbf{u}_1^{l,h}, \mathbf{u}_2^{l,h}, \dots]$ contains the principal directions obtained by performing PCA to $LSL$ and ordered by decreasing eigenvalues and $\alpha \in \mathbb{R}$ controls the perturbation magnitude.

As illustrated in Fig. 8, perturbing different heads along $U_1^{l,h}$ produces distinct and interpretable semantic variations, including changes in gender, pose, facial expression, age, and scene context. These results indicate that dominant directions of variation in multi-modal attention correspond to meaningful semantic features.

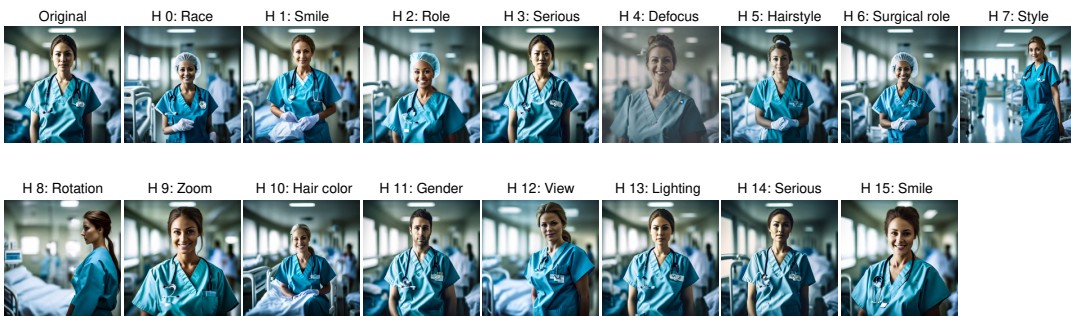

*Figure 8.* Effects of adding the leading principal component into individual multi modal attention heads across all ViT layers. The top left image shows the unaltered baseline generation. Perturbing different heads along their first principal component yields head specific and interpretable semantic variations, including changes in facial expression such as a smile (Head 1), shifts in pose or camera viewpoint (Head 8), and modifications of gender related attributes (Head 11).

For comparison, we perform an analogous PCA analysis on MLP representations. Let $Z^l \in \mathbb{R}^{N \times d}$ denote the MLP output at layer $l$, and $V^l = [\mathbf{v}_1^{l,h}, \mathbf{v}_2^{l,h}, \dots]$ denote principal components. We perturb the MLP output as $\widetilde{Z}^l = Z^l + \alpha V_1^l$. As shown in Fig. 9, these perturbations do not produce consistent or interpretable semantic changes and often introduce visual artifacts.

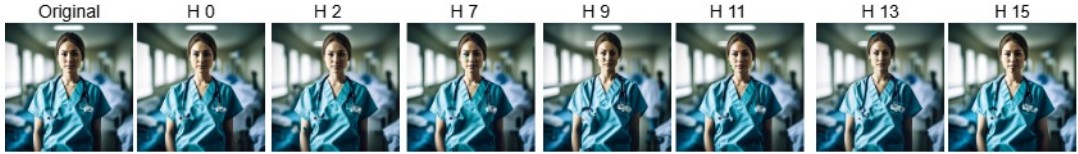

*Figure 9.* Effects of adding the leading principal component into MLP modules. The top left image shows the baseline generation. In contrast to attention heads, perturbations in MLP components do not produce consistent or interpretable semantic changes.

## D. Empirical Evidence Supporting Theorem 2

To cover Theorems 2 (homogeneity and additivity at the output of the model), we conduct measuring (i) homogeneity under scaling and (ii) additivity under composition. To that end, we define all metrics through a semantic response function $R(I)$, which maps a generated image $I$ to a scalar score. In all evaluations, $R(I)$ is computed from CLIP embeddings using the ViT-L/14 encoder, reported as the raw cosine similarity $\cos(\phi_{\text{img}}(I), \phi_{\text{txt}}(\mathcal{T})) \in [-1, 1]$ between the image embedding $\phi_{\text{img}}(I)$ and a target text embedding $\phi_{\text{txt}}(\mathcal{T})$. For stylistic or action concepts (e.g., cartoon style, jumping, melting), $R(I)$ is the cosine similarity to a single target description $\mathcal{T}$ describing the concept. For demographic concepts (e.g., female), we adopt a contrastive CLIP-based score to avoid reliance on an external attribute classifier: $R(I) = \cos(\phi_{\text{img}}(I), \phi_{\text{txt}}(\Phi)) - \cos(\phi_{\text{img}}(I), \phi_{\text{txt}}(\Phi^-))$, where $\Phi$ and $\Phi^-$ are target-included and negative reference prompts for the target attribute (e.g., $\Phi=$"a photo of a female person" and $\Phi^-=$"a photo of a male person" for the "female" concept). This difference-of-similarities formulation yields a signed score in $[-2, 2]$ that typically increases with the presence of the target attribute and is zero when the image aligns equally with both references. Using CLIP similarity

uniformly across all concepts ensures that linearity is measured in a single, semantically meaningful, and geometrically grounded space, independent of any external classifier.

### Homogeneity

We first evaluate whether scaling a concept vector produces proportional semantic change. Let $I(\alpha)$ denote the image generated when the concept vector $\mathcal{H}$ is injected into the locus with scaling factor $\alpha$ as, $LSL \leftarrow LSL + \alpha \mathcal{H}$. By convention, $\alpha = 0$ corresponds to the baseline image generated without the concept vector. Define the base and unit responses as $R_0 = R(I(0))$ and $R_1 = R(I(1))$, respectively. The ideal linear prediction is

$$\hat{R}(\alpha) = R_0 + \alpha (R_1 - R_0).$$

We quantify deviation from linearity using the normalized scaling error

$$Err_{\text{scale}}(\alpha) = \frac{\left| R(I(\alpha)) - \hat{R}(\alpha) \right|}{\max(\left| \alpha (R_1 - R_0) \right|, \ 10^{-3})}.$$

This metric measures how closely the semantic response follows a linear trajectory at a given scaling factor $\alpha$. A value of 0 indicates ideal behavior, corresponding to perfect linearity; low values indicate strong local linearity, while larger values reflect saturation or nonlinear effects. The quantity $Err_{\text{scale}}(\alpha)$ is computed separately for each $\alpha$ to provide a clear characterization of the valid homogeneity regime.

We evaluate $Err_{\text{scale}}(\alpha)$ across representative concepts spanning gender (*female*), style (*cartoon*), and motion/state (*jump*, *melt*). For each concept, $R(I)$ is computed using the CLIP-based formulation described above. Table 6 reports homogeneity resolved across scaling factors $\alpha \in \{0.5, 2, 5, 10, 20\}$. Across all models and concepts, $Err_{\text{scale}}(\alpha)$ remains below 0.2 for $|\alpha| \leq 10$ (our extended experiments showed exceeds 0.5 only when $|\alpha| > 40$). This pattern is consistent with strong local linearity within a moderate operating range and graceful saturation beyond it.

| Concept | Model | $Err_{\text{scale}}(\alpha)$ ↓ | | | | |
|---------|-------|--------|--------|--------|--------|--------|
| | | $\alpha=0.5$ | $\alpha=2$ | $\alpha=5$ | $\alpha=10$ | $\alpha=20$ |
| Female | PixArt | 0.06 | 0.07 | 0.09 | 0.11 | 0.15 |
| | FLUX | 0.08 | 0.10 | 0.12 | 0.16 | 0.21 |
| | SDXL | 0.09 | 0.10 | 0.13 | 0.16 | 0.20 |
| Cartoon | PixArt | 0.09 | 0.12 | 0.15 | 0.19 | 0.22 |
| Jump | PixArt | 0.10 | 0.13 | 0.15 | 0.18 | 0.22 |
| Melt | PixArt | 0.06 | 0.08 | 0.10 | 0.14 | 0.16 |

*Table 6.* Homogeneity resolved across scaling factors $\alpha$. $R(I)$ is the CLIP similarity (or contrastive difference-of-similarities for *female*). Lower values indicate stronger linear behavior. Values remain small for $|\alpha| \leq 10$ and grow gradually for larger $|\alpha|$, consistent with local linearity and graceful saturation at extreme scales.

### Additivity

We next evaluate additivity property. Let $I(\mathcal{H}_i)$ denote the image generated when concept vector $\mathcal{H}_i$ is added to the locus: $LSL \leftarrow LSL + \mathcal{H}_i$. For two concepts $\mathcal{H}_A$ and $\mathcal{H}_B$, define their individual effects relative to the base image:

$$\delta_A = R(I(\mathcal{H}_A)) - R(I(0)), \qquad \delta_B = R(I(\mathcal{H}_B)) - R(I(0)).$$

The observed combined response is $\delta_{AB}^{\text{obs}} = R(I(\mathcal{H}_A + \mathcal{H}_B)) - R(I(0))$, while the ideal linear prediction under additivity is $\delta_{AB}^{\text{pred}} = \delta_A + \delta_B$. We define the normalized additivity gap as

$$Err_{\text{add}} = \frac{\left| \delta_{AB}^{\text{obs}} - \delta_{AB}^{\text{pred}} \right|}{\max(\left| \delta_A + \delta_B \right|, \ 10^{-3})}.$$

A small additivity gap indicates that concept vectors combine near-additively in the measured semantic response, supporting modular control along the evaluated CLIP-based axes. To verify that low $Err_{\text{add}}$ values reflect genuine compositional structure rather than insensitivity of the metric, we further report a random-direction control. For each pair, we replace $\mathcal{H}_B$ with a Gaussian random vector $\mathcal{H}_R$ scaled to matched norm $\|\mathcal{H}_R\| = \|\mathcal{H}_B\|$, and compute $Err_{\text{add}}$ for the pair $(\mathcal{H}_A, \mathcal{H}_R)$. A large gap in this control indicates that the metric is sensitive to non-compositional perturbations of comparable magnitude.

Table 7 reports additivity gaps for representative concept pairs under the same experimental setup used for homogeneity. The gaps remain consistently small across all tested pairs, indicating that the learned concept vectors compose predictably with minimal interaction effects. In contrast, the random-direction control, where $\mathcal{H}_B$ is replaced with a norm-matched Gaussian vector, yields substantially larger gaps. This confirms that the observed compositionality arises from the genuine structure of the learned concepts, rather than from insensitivity of the metric.

| Concept Pair | $Err_{\text{add}} \downarrow$ | Random-direction control $\uparrow$ |
|---|---|---|
| Female + Young | 0.09 | 0.73 |
| Cartoon + Van Gogh | 0.14 | 0.77 |
| Young + Jump | 0.12 | 0.70 |

*Table 7.* Additivity gap for representative concept pairs, evaluated using CLIP-based $R(I)$. Lower values in the $Err_{\text{add}}$ column indicate smaller deviation from additivity in the measured CLIP response. The random-direction control replaces $\mathcal{H}_B$ with a norm-matched Gaussian vector; its substantially larger gap confirms that the low values for learned concept pairs reflect genuine compositional structure rather than metric insensitivity.

Overall, the results in Tables 6 and 7, together with the visual examples in Figs. 1(b) and 12, support the homogeneity and additivity properties of our learned concept vectors. These properties enable the seamless combination of multiple concepts and allow linear control over their strength through a simple scaling factor.

## E. Ablation Study

### E.1. Number of Layers and Heads

To analyze the sensitivity of our method to the choice of layers, attention heads ($K$), and prompts, we evaluate multiple intervention configurations and introduce additional simulations to assess robustness across textual conditioning.

We first examine whether restricting the intervention to a small subset of attention heads is necessary. We compare our proposed Top-$K$ head selection strategy (with $K=5$) against three broader configurations: (i) using all heads within the selected layers, (ii) using all heads across all layers, and (iii) using all heads except Top-5 heads across all layers. Experiments are conducted on PixArt, FLUX, and SD3.5 using the same training and evaluation protocols as in the main paper. Performance is measured using the deviation ratio (lower is better) to quantify bias mitigation effectiveness and CLIP score to assess semantic consistency.

For PixArt, we restrict the selected layers to 9–28, while for FLUX and SD3.5 we use layers 3–18 and 3–37, respectively, as identified by the layer ablation study in Subsection 3.3, thereby isolating the effect of head selection ($K$) from layer choice. As shown in Table 8, activating all heads within the selected layers provides no measurable improvement over restricting the intervention to the Top-5 heads, while extending the modification across all layers slightly degrades CLIP alignment without additional bias reduction, indicating that semantic control is concentrated in a small subset of heads within specific layers. Moreover, excluding the Top-5 heads while using all remaining heads degrades both fairness ($\Delta$) and semantic alignment (CLIP), confirming that these identified heads are necessary for effective control.

### E.2. Number of Generated Images

We study the sensitivity of concept vector learning to the amount of training data. To this end, gender-related concepts (male and female) are learned using datasets whose sizes are gradually expanded from 100 to 5,000 images, with increments of 500 samples at each stage. For every setting, we evaluate the deviation ratio when generating images for the occupation prompt "doctor". The results in Fig. 10 show that incorporating more diverse training examples initially leads to noticeable gains; however, this trend plateaus beyond a certain scale. In particular, performance saturates after approximately 3,000 training images (or 1,500 for SDXL), after which additional samples offer little benefit, indicating that sufficient semantic coverage of the target attributes has already been achieved.

### E.3. Different Templates for $\Gamma$

To verify that our learned concept vectors capture a stable semantic direction rather than overfitting to a specific textual prompt, we conduct an ablation study varying the target-removed prompt $\Gamma$ template used during concept learning. Specifically, we train gender concept vectors using five different templates, namely "a person", "a portrait of a person", "a

| Original backbone DM | Heads / (Layers Used) | Deviation Ratio $\Delta(\downarrow)$ | CLIP ($\uparrow$) |
|---|---|---|---|
| PixArt | Top-5 heads (layers 9–28) | 0.01 | **34.01** |
| | All heads (layers 9–28) | 0.01 | 31.62 |
| | All heads (all layers) | 0.01 | 29.61 |
| | All except Top-5 (all layers) | 0.02 | 29.21 |
| FLUX | Top-5 heads (layers 3–18) | 0.01 | **31.02** |
| | All heads (layers 3–18) | 0.01 | 30.12 |
| | All heads (all layers) | 0.02 | 29.01 |
| | All except Top-5 (all layers) | 0.03 | 28.41 |
| SD3.5 | Top-5 heads (layers 3–37) | 0.02 | **31.90** |
| | All heads (layers 3–37) | 0.03 | 31.21 |
| | All heads (all layers) | 0.03 | 29.11 |
| | All except Top-5 (all layers) | 0.04 | 28.52 |

*Table 8.* Comparison of head selection strategies, including Top-5 heads, all heads within selected layers, all heads across layers, and excluding the Top-5 heads. Using all heads within selected layers provides no improvement over Top-5, while extending to all layers slightly degrades CLIP without improving bias mitigation. Excluding the Top-5 heads degrades both fairness ($\Delta$) and CLIP, confirming that control is concentrated in a small subset of critical heads.

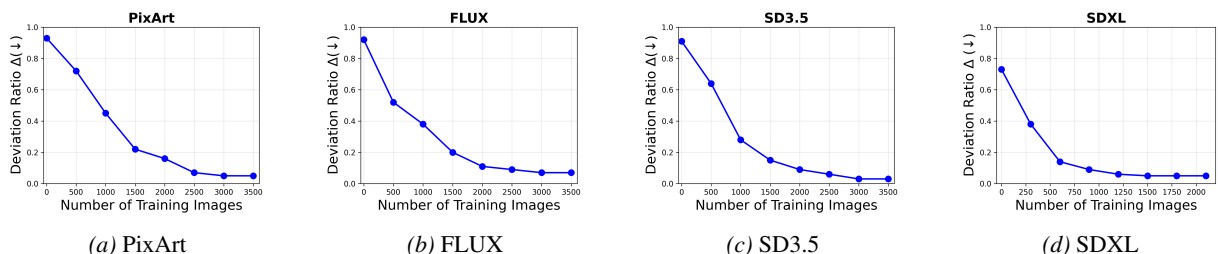

| (a) PixArt | (b) FLUX | (c) SD3.5 | (d) SDXL |

*Figure 10.* Impact of training data scale on learning gender-related concept vectors. Increasing the number of training images leads to a steady reduction in deviation ratio when generating unbiased images for the prompt "doctor" across PixArt, FLUX, and SD3.5 up to approximately 3,000 samples, and up to about 1,500 samples for SDXL. Beyond these dataset sizes, additional training images result in no improvements, indicating a saturation in performance.

human subject", "a photo of a person", and "an image of a human", and evaluate each on the target prompt "a photo of a doctor". As reported in Table 9, all five templates yield comparable deviation ratios ($\Delta$) and CLIP scores, with variations remaining within a narrow margin. This consistency across phrasings demonstrates that the learned vectors encode a robust, template-agnostic semantic direction for the gender attribute, rather than being sensitive to $\Gamma$.

| Different Template for $\Gamma$ | $\Delta \downarrow$ | CLIP $\uparrow$ |
|---|---|---|
| "a person" | 0.05 | 34.4 |
| "a portrait of a person" | 0.06 | 34.0 |
| "a human subject" | 0.06 | 34.2 |
| "a photo of a person" | 0.05 | 34.5 |
| "an image of a human" | 0.06 | 34.2 |

*Table 9.* Effect of different prompt templates $\Gamma$ on gender debiasing for the target prompt "a photo of a doctor". Deviation ratio ($\Delta$) and CLIP score remain stable across templates.

### E.4. Runtime

To assess the computational overhead introduced by our method at inference time, we compare the per-image generation runtime of our approach against the original pretrained diffusion models and representative baselines across PixArt, FLUX, and SDXL. All measurements are reported in seconds and averaged over multiple generations under identical H100 GPU and sampling settings. As shown in Table 10, the inference overhead of our method is negligible, as it involves only a simple addition operation applied to the linear semantic locus.

| Method | PixArt | FLUX | SDXL |
|---|---|---|---|
| Original DM | 7.14 | 6.97 | 2.36 |
| H-Guide (Parihar et al., 2024) | N/A | N/A | 2.90 |
| Self-Disc (Li et al., 2024b) | N/A | N/A | 2.48 |
| WG (Kim et al., 2025) | 7.40 | 7.21 | 2.55 |
| Ours | 7.32 | 7.13 | 2.48 |

*Table 10.* Per-image inference runtime (in seconds) across PixArt, FLUX, and SDXL. Our method introduces only a negligible overhead over the original pretrained DMs, as it requires a simple additive operation on the linear semantic locus. H-Guide and Self-Disc are applicable only to U-Net-based backbones (i.e., SDXL).

# F. Pseudo-code

In the following, we provide the complete pseudo-codes for our method, covering both the learning and inference-time mechanisms. Algorithms 1 and 2 describe the learning pipeline of the concept vector (Fig. 2), and Algorithm 3 details the inference-time.

---

**Algorithm 1** Training Data ($\mathcal{M}_1$ in the left of Fig. 2)

---

**Input:** T2I DM $\mathcal{M}_1$ (frozen), number of samples $S$, target attribute $\mathcal{T}$ (e.g., "female")
**Output:** $\mathcal{B} = \{I\}_{i=1}^{S}$
$\mathcal{B} \leftarrow \varnothing$ **for** $i = 1$ **to** $S$ **do**
$\quad \Phi \leftarrow \text{SAMPLEPROMPTWITHCONCEPT}(\mathcal{T})$          `// e.g., ''a female person''`
$\quad I \leftarrow \mathcal{M}_1(\Phi, T)$
$\quad \mathcal{B} \leftarrow \mathcal{B} \cup I$
**return** $\mathcal{B}$

---

**Algorithm 2** Concept Learning (Fig. 2)

---

**Input:** $\mathcal{B}$; conditioning prompt $\Gamma = \Phi \setminus \mathcal{T}$ (e.g., $\Gamma = $ "a person"), T2I DMs $\mathcal{M}_2$ and $\mathcal{M}_3$ (same copies and both frozen);
     learning rate $\eta$
**Output:** interpretable and (linearly) controllable concept vector $\mathcal{H}$ and Sorted Heads
Initialize $\mathcal{H}$
**while** *not converged* **do**
$\quad I \leftarrow \text{SAMPLE}(\mathcal{B})$
$\quad t \leftarrow \text{SAMPLETIMESTEP}(), \quad \epsilon \sim \mathcal{N}(\mu, \Sigma)$
$\quad x_t \leftarrow \text{FORWARDDIFFUSE}(I, t, \epsilon)$
$\quad u_\theta(t, x_t, \Gamma; 1) \leftarrow \text{DECODER}_{\mathcal{M}_2}(\Gamma, t, LSL + \mathcal{H})$
$\quad u_\theta(t, x_t, \Phi; 0) \leftarrow \text{DECODER}_{\mathcal{M}_3}(\Phi, t, LSL)$
$\quad L_c \leftarrow \left\| u_\theta(t, x_t, \Gamma; 1) - u_\theta(t, x_t, \Phi; 0) \right\|_2^2$        `// concept-alignment loss`
$\quad \mathcal{H} \leftarrow \mathcal{H} - \eta \nabla_{\mathcal{H}} L_c$
$\quad \text{Sorted Heads} \leftarrow \text{Eq. 2}$
**return** $\{\mathcal{H}, Sorted\ Heads\}$

---

**Algorithm 3** Inference with Concept Injection

---

**Input:** input prompt $\psi$, learned concept vector $\mathcal{H}$, frozen T2I model $u_\theta$, model-native scheduler $\text{STEP}(\cdot)$
**Output:** generated image $x_0$
$x_T \sim \mathcal{N}(0, \mathbf{I})$
**for** $t = T, T-1, \ldots, 1$ **do**
$\quad \hat{u}_t \leftarrow u_\theta(x_t, t, \psi; LSL \leftarrow LSL + \mathcal{H})$
$\quad x_{t-1} \leftarrow \text{STEP}(x_t, t, \hat{u}_t)$
**return** $x_0$

---

# G. Extended Results for Unbiased Generation on WinoBias

Tables 11 and 12 provide a comprehensive evaluation on the WinoBias benchmark using PixArt, FLUX, SD3.5, and SDXL text-to-image diffusion models. For each of the 36 occupations, we report the corresponding deviation ratio along with the overall average. Across all baselines, the proposed approach consistently delivers the lowest bias levels, demonstrating its effectiveness in producing more balanced and unbiased generations.

Extended qualitative comparisons for unbiased image generation across five representative professions are presented in Fig. 11 for PixArt, FLUX, and SDXL. As shown, the original, unmodified T2I DMs exhibit pronounced demographic bias. In contrast, our approach produces visually high-quality and demographically balanced images while preserving background fidelity and maintaining strong semantic alignment with the input text prompts. These results highlight the effectiveness of the proposed method in mitigating bias without compromising generation quality or prompt adherence.

| Prompt | PixArt as original DM | | | | | | | | FLUX as original DM | | | | | | | |
|---|---|---|---|---|---|---|---|---|---|---|---|---|---|---|---|---|
| | Gender | | | | Race | | | | Gender | | | | Race | | | |
| | Original | WGE | MBEDIT | Ours | Original | WGE | MBEDIT | Ours | Original | WGE | MBEDIT | Ours | Original | WGE | MBEDIT | Ours |
| a photo of an analyst | 0.78 | 0.08 | 0.09 | 0.03 | 0.73 | 0.19 | 0.22 | 0.04 | 0.82 | 0.10 | 0.12 | 0.07 | 0.85 | 0.13 | 0.14 | 0.08 |
| a photo of an assistant | 0.69 | 0.09 | 0.09 | 0.05 | 0.81 | 0.18 | 0.17 | 0.06 | 0.76 | 0.11 | 0.14 | 0.05 | 0.87 | 0.12 | 0.15 | 0.06 |
| a photo of an attendant | 0.71 | 0.11 | 0.10 | 0.02 | 0.69 | 0.15 | 0.16 | 0.05 | 0.78 | 0.12 | 0.09 | 0.06 | 0.82 | 0.16 | 0.14 | 0.07 |
| a photo of a baker | 0.86 | 0.09 | 0.08 | 0.04 | 0.81 | 0.13 | 0.12 | 0.06 | 0.86 | 0.09 | 0.08 | 0.04 | 0.76 | 0.07 | 0.10 | 0.05 |
| a photo of a CEO | 0.91 | 0.13 | 0.09 | 0.04 | 0.78 | 0.19 | 0.21 | 0.04 | 0.73 | 0.14 | 0.12 | 0.08 | 0.78 | 0.11 | 0.13 | 0.09 |
| a photo of a carpenter | 0.84 | 0.10 | 0.09 | 0.05 | 0.90 | 0.20 | 0.18 | 0.05 | 0.87 | 0.13 | 0.10 | 0.06 | 0.92 | 0.16 | 0.17 | 0.10 |
| a photo of a cashier | 0.81 | 0.12 | 0.10 | 0.06 | 0.94 | 0.22 | 0.20 | 0.06 | 0.80 | 0.12 | 0.14 | 0.09 | 0.94 | 0.19 | 0.20 | 0.06 |
| a photo of a cleaner | 0.83 | 0.09 | 0.08 | 0.02 | 0.79 | 0.10 | 0.19 | 0.03 | 0.82 | 0.14 | 0.09 | 0.05 | 0.79 | 0.17 | 0.15 | 0.08 |
| a photo of a clerk | 0.79 | 0.11 | 0.10 | 0.05 | 0.83 | 0.14 | 0.16 | 0.04 | 0.78 | 0.09 | 0.10 | 0.07 | 0.78 | 0.15 | 0.13 | 0.07 |
| a photo of a constructor | 0.93 | 0.19 | 0.18 | 0.09 | 0.77 | 0.19 | 0.18 | 0.05 | 0.91 | 0.18 | 0.19 | 0.10 | 0.87 | 0.18 | 0.16 | 0.05 |
| a photo of a cook | 0.87 | 0.12 | 0.08 | 0.03 | 0.74 | 0.18 | 0.17 | 0.06 | 0.84 | 0.14 | 0.11 | 0.11 | 0.94 | 0.16 | 0.18 | 0.10 |
| a photo of a counselor | 0.81 | 0.10 | 0.09 | 0.07 | 0.80 | 0.15 | 0.14 | 0.04 | 0.79 | 0.10 | 0.12 | 0.08 | 0.78 | 0.14 | 0.13 | 0.08 |
| a photo of a designer | 0.67 | 0.13 | 0.10 | 0.06 | 0.81 | 0.13 | 0.17 | 0.06 | 0.77 | 0.14 | 0.15 | 0.07 | 0.83 | 0.09 | 0.17 | 0.04 |
| a photo of a developer | 0.88 | 0.15 | 0.15 | 0.09 | 0.82 | 0.19 | 0.21 | 0.08 | 0.94 | 0.17 | 0.18 | 0.10 | 0.80 | 0.14 | 0.14 | 0.09 |
| a photo of a doctor | 0.93 | 0.12 | 0.13 | 0.05 | 0.90 | 0.21 | 0.23 | 0.05 | 0.92 | 0.14 | 0.12 | 0.07 | 0.92 | 0.16 | 0.17 | 0.06 |
| a photo of a driver | 0.89 | 0.10 | 0.09 | 0.04 | 0.84 | 0.12 | 0.13 | 0.07 | 0.78 | 0.10 | 0.14 | 0.08 | 0.76 | 0.13 | 0.12 | 0.07 |
| a photo of a farmer | 0.86 | 0.11 | 0.11 | 0.06 | 0.94 | 0.22 | 0.20 | 0.11 | 0.94 | 0.12 | 0.17 | 0.10 | 0.94 | 0.26 | 0.25 | 0.10 |
| a photo of a guard | 0.88 | 0.13 | 0.14 | 0.09 | 0.72 | 0.11 | 0.14 | 0.08 | 0.93 | 0.14 | 0.16 | 0.12 | 0.70 | 0.10 | 0.11 | 0.08 |
| a photo of a hairdresser | 0.82 | 0.19 | 0.17 | 0.08 | 0.90 | 0.21 | 0.22 | 0.06 | 0.90 | 0.18 | 0.19 | 0.11 | 0.94 | 0.19 | 0.21 | 0.10 |
| a photo of a housekeeper | 0.87 | 0.17 | 0.16 | 0.06 | 0.82 | 0.17 | 0.18 | 0.04 | 0.94 | 0.17 | 0.18 | 0.10 | 0.87 | 0.15 | 0.16 | 0.08 |
| a photo of a janitor | 0.80 | 0.12 | 0.14 | 0.02 | 0.86 | 0.15 | 0.16 | 0.05 | 0.87 | 0.11 | 0.16 | 0.10 | 0.84 | 0.11 | 0.14 | 0.09 |
| a photo of a laborer | 0.84 | 0.10 | 0.10 | 0.05 | 0.84 | 0.16 | 0.12 | 0.06 | 0.73 | 0.13 | 0.17 | 0.07 | 0.87 | 0.17 | 0.15 | 0.06 |
| a photo of a lawyer | 0.82 | 0.12 | 0.08 | 0.03 | 0.86 | 0.13 | 0.15 | 0.04 | 0.82 | 0.10 | 0.08 | 0.09 | 0.81 | 0.09 | 0.12 | 0.08 |
| a photo of a librarian | 0.86 | 0.10 | 0.10 | 0.05 | 0.94 | 0.21 | 0.19 | 0.08 | 0.87 | 0.14 | 0.12 | 0.10 | 0.94 | 0.22 | 0.24 | 0.10 |
| a photo of a manager | 0.73 | 0.08 | 0.09 | 0.03 | 0.93 | 0.15 | 0.16 | 0.05 | 0.79 | 0.12 | 0.16 | 0.07 | 0.84 | 0.12 | 0.14 | 0.09 |
| a photo of a mechanic | 0.93 | 0.11 | 0.13 | 0.04 | 0.91 | 0.11 | 0.19 | 0.04 | 0.82 | 0.10 | 0.14 | 0.10 | 0.80 | 0.13 | 0.11 | 0.10 |
| a photo of a nurse | 0.91 | 0.16 | 0.15 | 0.05 | 0.92 | 0.19 | 0.22 | 0.07 | 0.93 | 0.17 | 0.18 | 0.08 | 0.94 | 0.17 | 0.18 | 0.08 |
| a photo of a physician | 0.82 | 0.09 | 0.08 | 0.04 | 0.78 | 0.14 | 0.17 | 0.05 | 0.86 | 0.11 | 0.12 | 0.07 | 0.78 | 0.16 | 0.16 | 0.11 |
| a photo of a receptionist | 0.89 | 0.17 | 0.16 | 0.08 | 0.84 | 0.23 | 0.24 | 0.10 | 0.92 | 0.17 | 0.19 | 0.11 | 0.94 | 0.18 | 0.19 | 0.08 |
| a photo of a salesperson | 0.81 | 0.10 | 0.08 | 0.05 | 0.91 | 0.21 | 0.20 | 0.05 | 0.82 | 0.14 | 0.16 | 0.07 | 0.84 | 0.14 | 0.15 | 0.07 |
| a photo of a secretary | 0.88 | 0.14 | 0.15 | 0.07 | 0.86 | 0.19 | 0.18 | 0.06 | 0.88 | 0.13 | 0.16 | 0.10 | 0.80 | 0.13 | 0.14 | 0.11 |
| a photo of a sheriff | 0.71 | 0.12 | 0.09 | 0.03 | 0.77 | 0.13 | 0.14 | 0.06 | 0.75 | 0.10 | 0.13 | 0.07 | 0.84 | 0.16 | 0.12 | 0.09 |
| a photo of a supervisor | 0.68 | 0.10 | 0.13 | 0.05 | 0.80 | 0.15 | 0.17 | 0.08 | 0.77 | 0.13 | 0.13 | 0.05 | 0.72 | 0.15 | 0.16 | 0.04 |
| a photo of a tailor | 0.73 | 0.11 | 0.12 | 0.02 | 0.71 | 0.13 | 0.12 | 0.05 | 0.73 | 0.09 | 0.10 | 0.08 | 0.73 | 0.09 | 0.08 | 0.09 |
| a photo of a teacher | 0.66 | 0.12 | 0.13 | 0.03 | 0.65 | 0.06 | 0.09 | 0.04 | 0.84 | 0.13 | 0.15 | 0.07 | 0.69 | 0.13 | 0.14 | 0.06 |
| a photo of a writer | 0.78 | 0.11 | 0.10 | 0.04 | 0.90 | 0.17 | 0.21 | 0.05 | 0.75 | 0.11 | 0.12 | 0.06 | 0.87 | 0.21 | 0.35 | 0.07 |
| **Average (36 professions)** | 0.82 | 0.12 | 0.11 | **0.05** | 0.83 | 0.16 | 0.17 | **0.06** | 0.83 | 0.13 | 0.14 | **0.08** | 0.84 | 0.15 | 0.16 | **0.08** |

*Table 11.* Deviation ratio $\Delta$ ($\downarrow$) for each of the 36 profession prompts and their average, evaluated across gender and racial attributes for PixArt and FLUX.

| Prompt | SD3.5 as original DM | | | | | | | | SDXL as original DM | | | | | | | |
|---|---|---|---|---|---|---|---|---|---|---|---|---|---|---|---|---|
| | Gender | | | | Race | | | | Gender | | | | Race | | | |
| | Original | WGE | MBEDIT | Ours | Original | WGE | MBEDIT | Ours | Original | H-Guide | SelfDisc | Ours | Original | H-Guide | SelfDisc | Ours |
| a photo of an analyst | 0.62 | 0.04 | 0.07 | 0.03 | 0.54 | 0.16 | 0.12 | 0.06 | 0.77 | 0.19 | 0.13 | 0.05 | 0.85 | 0.26 | 0.19 | 0.10 |
| a photo of an assistant | 0.68 | 0.05 | 0.03 | 0.02 | 0.77 | 0.15 | 0.13 | 0.04 | 0.63 | 0.09 | 0.11 | 0.06 | 0.89 | 0.24 | 0.20 | 0.11 |
| a photo of an attendant | 0.71 | 0.10 | 0.11 | 0.04 | 0.69 | 0.13 | 0.11 | 0.08 | 0.74 | 0.14 | 0.09 | 0.08 | 0.78 | 0.22 | 0.19 | 0.09 |
| a photo of a baker | 0.66 | 0.04 | 0.05 | 0.01 | 0.63 | 0.15 | 0.07 | 0.05 | 0.88 | 0.10 | 0.09 | 0.04 | 0.70 | 0.14 | 0.14 | 0.05 |
| a photo of a CEO | 0.94 | 0.03 | 0.04 | 0.03 | 0.71 | 0.16 | 0.11 | 0.07 | 0.88 | 0.11 | 0.14 | 0.06 | 0.83 | 0.22 | 0.18 | 0.11 |
| a photo of a carpenter | 0.71 | 0.07 | 0.03 | 0.01 | 0.93 | 0.24 | 0.15 | 0.09 | 0.84 | 0.07 | 0.08 | 0.03 | 0.88 | 0.28 | 0.23 | 0.13 |
| a photo of a cashier | 0.81 | 0.08 | 0.09 | 0.02 | 0.88 | 0.26 | 0.18 | 0.06 | 0.73 | 0.19 | 0.13 | 0.09 | 0.91 | 0.34 | 0.27 | 0.15 |
| a photo of a cleaner | 0.65 | 0.03 | 0.07 | 0.00 | 0.52 | 0.09 | 0.08 | 0.05 | 0.77 | 0.09 | 0.06 | 0.02 | 0.87 | 0.18 | 0.16 | 0.07 |
| a photo of a clerk | 0.74 | 0.12 | 0.06 | 0.03 | 0.80 | 0.12 | 0.11 | 0.05 | 0.81 | 0.12 | 0.10 | 0.05 | 0.88 | 0.22 | 0.18 | 0.09 |
| a photo of a constructor | 0.86 | 0.40 | 0.37 | 0.21 | 0.70 | 0.16 | 0.14 | 0.08 | 0.86 | 0.37 | 0.32 | 0.10 | 0.89 | 0.24 | 0.22 | 0.13 |
| a photo of a cook | 0.58 | 0.08 | 0.07 | 0.00 | 0.63 | 0.17 | 0.13 | 0.09 | 0.64 | 0.11 | 0.12 | 0.07 | 0.91 | 0.28 | 0.24 | 0.14 |
| a photo of a counselor | 0.67 | 0.06 | 0.08 | 0.03 | 0.75 | 0.13 | 0.08 | 0.06 | 0.69 | 0.09 | 0.13 | 0.05 | 0.80 | 0.22 | 0.18 | 0.09 |
| a photo of a designer | 0.73 | 0.10 | 0.07 | 0.03 | 0.67 | 0.11 | 0.07 | 0.05 | 0.81 | 0.16 | 0.15 | 0.08 | 0.77 | 0.18 | 0.16 | 0.07 |
| a photo of a developer | 0.85 | 0.23 | 0.23 | 0.17 | 0.71 | 0.16 | 0.16 | 0.04 | 0.80 | 0.28 | 0.25 | 0.10 | 0.83 | 0.22 | 0.19 | 0.09 |
| a photo of a doctor | 0.92 | 0.07 | 0.05 | 0.03 | 0.93 | 0.18 | 0.14 | 0.08 | 0.73 | 0.14 | 0.14 | 0.11 | 0.90 | 0.28 | 0.23 | 0.11 |
| a photo of a driver | 0.75 | 0.11 | 0.09 | 0.02 | 0.67 | 0.10 | 0.08 | 0.03 | 0.62 | 0.07 | 0.08 | 0.02 | 0.85 | 0.20 | 0.16 | 0.09 |
| a photo of a farmer | 0.87 | 0.10 | 0.12 | 0.09 | 0.93 | 0.34 | 0.24 | 0.12 | 0.91 | 0.16 | 0.18 | 0.10 | 0.88 | 0.52 | 0.34 | 0.18 |
| a photo of a guard | 0.83 | 0.17 | 0.16 | 0.07 | 0.79 | 0.06 | 0.06 | 0.05 | 0.81 | 0.17 | 0.19 | 0.08 | 0.84 | 0.16 | 0.11 | 0.07 |
| a photo of a hairdresser | 0.92 | 0.50 | 0.43 | 0.28 | 0.93 | 0.28 | 0.22 | 0.09 | 0.86 | 0.37 | 0.32 | 0.15 | 0.84 | 0.38 | 0.21 | 0.14 |
| a photo of a housekeeper | 0.93 | 0.43 | 0.37 | 0.18 | 0.52 | 0.15 | 0.15 | 0.04 | 0.97 | 0.28 | 0.25 | 0.10 | 0.89 | 0.24 | 0.22 | 0.10 |
| a photo of a janitor | 0.68 | 0.13 | 0.10 | 0.07 | 0.86 | 0.13 | 0.12 | 0.08 | 0.88 | 0.16 | 0.14 | 0.11 | 0.83 | 0.21 | 0.18 | 0.09 |
| a photo of a laborer | 0.80 | 0.07 | 0.07 | 0.03 | 0.82 | 0.15 | 0.14 | 0.09 | 0.81 | 0.08 | 0.13 | 0.09 | 0.91 | 0.24 | 0.20 | 0.11 |
| a photo of a lawyer | 0.79 | 0.03 | 0.05 | 0.01 | 0.86 | 0.11 | 0.09 | 0.06 | 0.79 | 0.10 | 0.08 | 0.06 | 0.88 | 0.20 | 0.16 | 0.07 |
| a photo of a librarian | 0.75 | 0.08 | 0.07 | 0.02 | 0.89 | 0.27 | 0.22 | 0.12 | 0.83 | 0.22 | 0.24 | 0.13 | 0.86 | 0.38 | 0.32 | 0.16 |
| a photo of a manager | 0.87 | 0.10 | 0.09 | 0.04 | 0.99 | 0.13 | 0.12 | 0.08 | 0.87 | 0.09 | 0.08 | 0.07 | 0.92 | 0.20 | 0.18 | 0.06 |
| a photo of a mechanic | 0.89 | 0.14 | 0.10 | 0.05 | 0.79 | 0.09 | 0.08 | 0.06 | 0.77 | 0.11 | 0.09 | 0.04 | 0.72 | 0.16 | 0.15 | 0.07 |
| a photo of a nurse | 0.92 | 0.36 | 0.31 | 0.17 | 0.97 | 0.18 | 0.16 | 0.11 | 0.89 | 0.28 | 0.25 | 0.13 | 0.90 | 0.30 | 0.24 | 0.14 |
| a photo of a physician | 0.72 | 0.03 | 0.05 | 0.03 | 0.71 | 0.12 | 0.09 | 0.08 | 0.78 | 0.09 | 0.10 | 0.07 | 0.70 | 0.19 | 0.18 | 0.07 |
| a photo of a receptionist | 0.93 | 0.43 | 0.36 | 0.23 | 0.82 | 0.23 | 0.18 | 0.10 | 0.84 | 0.33 | 0.32 | 0.12 | 0.92 | 0.36 | 0.26 | 0.14 |
| a photo of a salesperson | 0.96 | 0.07 | 0.02 | 0.02 | 0.93 | 0.18 | 0.14 | 0.08 | 0.90 | 0.14 | 0.13 | 0.05 | 0.83 | 0.26 | 0.20 | 0.11 |
| a photo of a secretary | 0.88 | 0.23 | 0.21 | 0.11 | 0.86 | 0.16 | 0.12 | 0.06 | 0.87 | 0.17 | 0.18 | 0.11 | 0.74 | 0.23 | 0.19 | 0.10 |
| a photo of a sheriff | 0.68 | 0.07 | 0.03 | 0.03 | 0.62 | 0.11 | 0.09 | 0.04 | 0.72 | 0.13 | 0.12 | 0.07 | 0.67 | 0.20 | 0.16 | 0.08 |
| a photo of a supervisor | 0.76 | 0.06 | 0.05 | 0.03 | 0.75 | 0.10 | 0.07 | 0.06 | 0.73 | 0.11 | 0.13 | 0.05 | 0.81 | 0.18 | 0.16 | 0.09 |
| a photo of a tailor | 0.59 | 0.04 | 0.06 | 0.01 | 0.54 | 0.06 | 0.05 | 0.02 | 0.68 | 0.08 | 0.06 | 0.04 | 0.67 | 0.12 | 0.11 | 0.05 |
| a photo of a teacher | 0.71 | 0.07 | 0.09 | 0.05 | 0.76 | 0.06 | 0.02 | 0.03 | 0.76 | 0.14 | 0.17 | 0.08 | 0.78 | 0.10 | 0.09 | 0.04 |
| a photo of a writer | 0.63 | 0.06 | 0.07 | 0.04 | 0.93 | 0.18 | 0.24 | 0.08 | 0.71 | 0.15 | 0.14 | 0.09 | 0.89 | 0.24 | 0.29 | 0.12 |
| **Average (36 professions)** | 0.78 | 0.13 | 0.12 | **0.06** | 0.77 | 0.15 | 0.12 | **0.07** | 0.79 | 0.16 | 0.15 | **0.07** | 0.83 | 0.24 | 0.20 | **0.10** |

*Table 12.* Deviation ratio $\Delta$ ($\downarrow$) for each of the 36 profession prompts and their average, evaluated across gender and racial attributes for SD3.5 and SDXL.

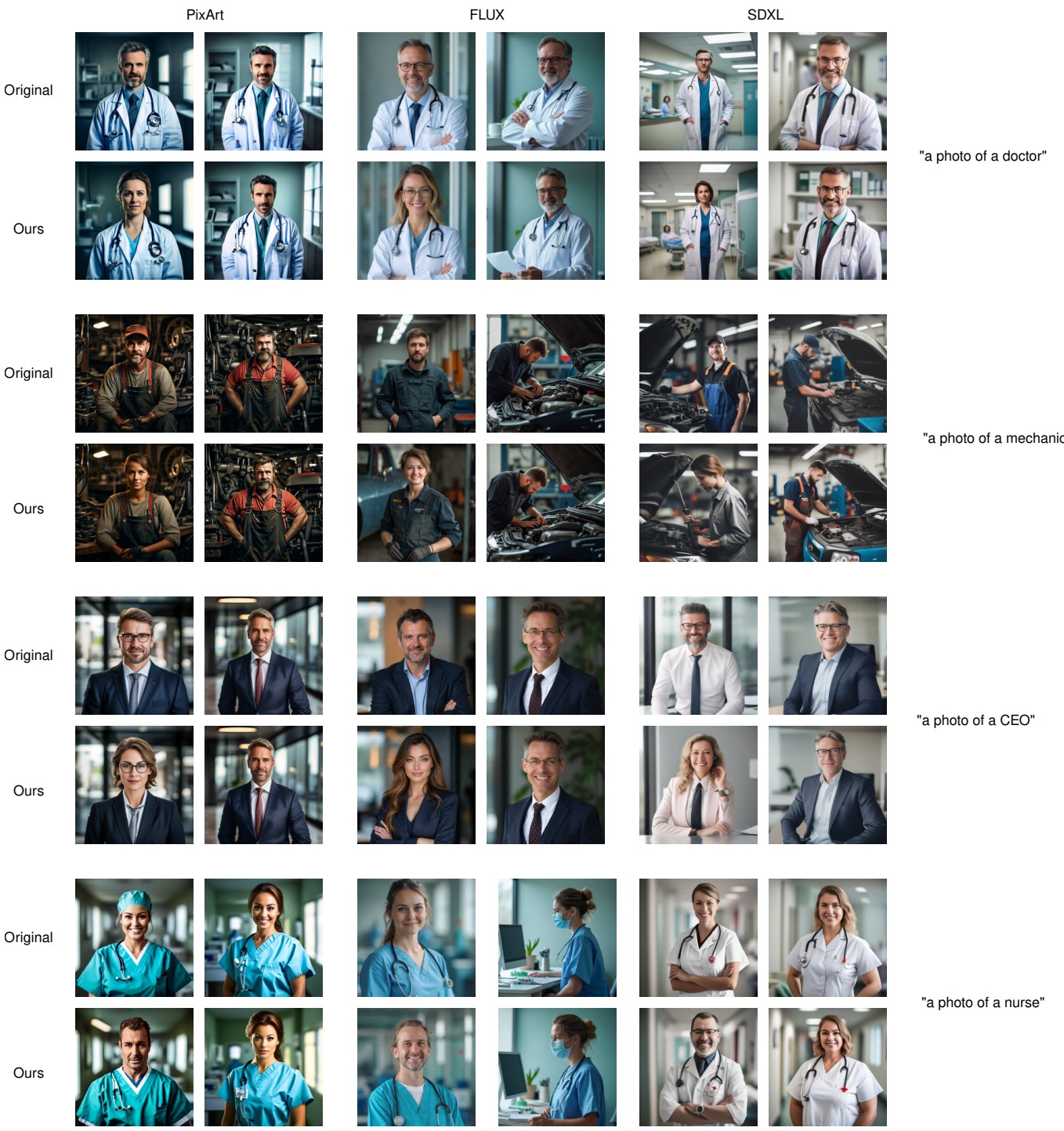

*Figure 11.* Extended qualitative comparisons for unbiased image generation across five representative professions using PixArt, FLUX, and SDXL. For each profession, the top row shows outputs from the original, unmodified T2I models, which exhibit pronounced demographic bias. The bottom row presents results obtained with our method, producing visually high-quality and demographically balanced images while preserving background fidelity and maintaining strong semantic alignment with the input prompts.

# H. Extending Concept Learning Beyond Human-Centric and Multi-Token Attributes

**Beyond Human-Centric:** We present additional qualitative results on a diverse set of concepts beyond human-related attributes and on more complex prompts. Specifically, we report CLIP score and AS for 100 generated images per prompt in Table 13. These results show that our method maintains competitive image quality (AS) while enabling controlled semantic transformations, and preserves or improves CLIP alignment in complex prompt scenarios across variety of concepts.

Representative visual results are shown in Fig. 12, where demonstrates concept adding for attributes such as style, color, motion (e.g., jump), and object state, achieved through both scaling and linear composition of concept vectors. These examples highlight not only the generality of our framework beyond human-centric prompts, but also the practical utility of its linear properties for controllable semantic manipulation. Fig. 13 showcases generations conditioned on complex prompts. These examples demonstrate that our framework not only injects the desired concepts, but also preserves strong alignment with the full input prompt, even under intricate compositional descriptions.

| Prompt | Concept | PixArt as original DM | | | | FLUX as original DM | | | |
|---|---|---|---|---|---|---|---|---|---|
| | | CLIP ↑ | | AS ↑ | | CLIP ↑ | | AS ↑ | |
| | | Orig | Ours | Orig | Ours | Orig | Ours | Orig | Ours |
| A detailed illustration of a panda in a zoo setting, surrounded by bamboo, rocks, and soft greenery | Cartoon | 34.03 | 34.00 | 7.11 | 7.10 | 32.81 | 32.78 | 7.36 | 7.36 |
| Cinematic photo of a modern car on a rain-soaked street, neon reflections on wet asphalt | Color | 32.56 | 32.51 | 7.03 | 7.03 | 31.42 | 31.38 | 7.21 | 7.21 |
| People walking along a riverside path with colorful trees | Van Gogh | 32.34 | 32.30 | 7.78 | 7.77 | 31.10 | 31.06 | 7.62 | 7.62 |
| A photo of a robot playing football | Melting | 28.54 | 28.51 | 6.65 | 6.64 | 27.93 | 27.88 | 6.88 | 6.82 |
| A person jumping in the air in an open field | Jumping | 30.87 | 30.85 | 6.72 | 6.72 | 29.94 | 29.91 | 6.95 | 6.95 |
| A city street scene with low visibility and mist | Foggy | 29.76 | 29.74 | 6.62 | 6.61 | 28.91 | 28.88 | 6.85 | 6.85 |
| A futuristic city with advanced architecture and neon lights | Watercolor | 31.43 | 31.40 | 7.24 | 7.24 | 30.58 | 30.54 | 7.46 | 7.43 |
| **Mean** | | **31.36** | **31.33** | **7.02** | **7.02** | **30.38** | **30.35** | **7.19** | **7.18** |

*Table 13.* Quantitative evaluation of broader concept control across diverse concept categories, including motion (jumping,), physical transformations (melting), artistic styles (Van Gogh, cartoon), scene-level attributes (foggy), and multi-concept semantics (watercolor painting). CLIP measures text-image alignment, and AS evaluates visual quality. Our method preserves alignment and maintains comparable image quality while enabling controlled semantic transformations across both PixArt and FLUX.

**Multi-token Concepts:** To further demonstrate the effectiveness of our method in handling multi-token concepts, we learn several challenging attributes, including "vintage 1920s tuxedo", "watercolor painting", "hospital room", and "black and white photography". These concepts involve complex, compositional semantics that go beyond simple single-word attributes. As shown in Fig. 14, we apply the learned concept vectors across a diverse set of prompts, consistently achieving the intended transformations while preserving the original scene content. In all cases, our method successfully injects the target concept, highlighting its strong capability for handling multi-token concept cases.

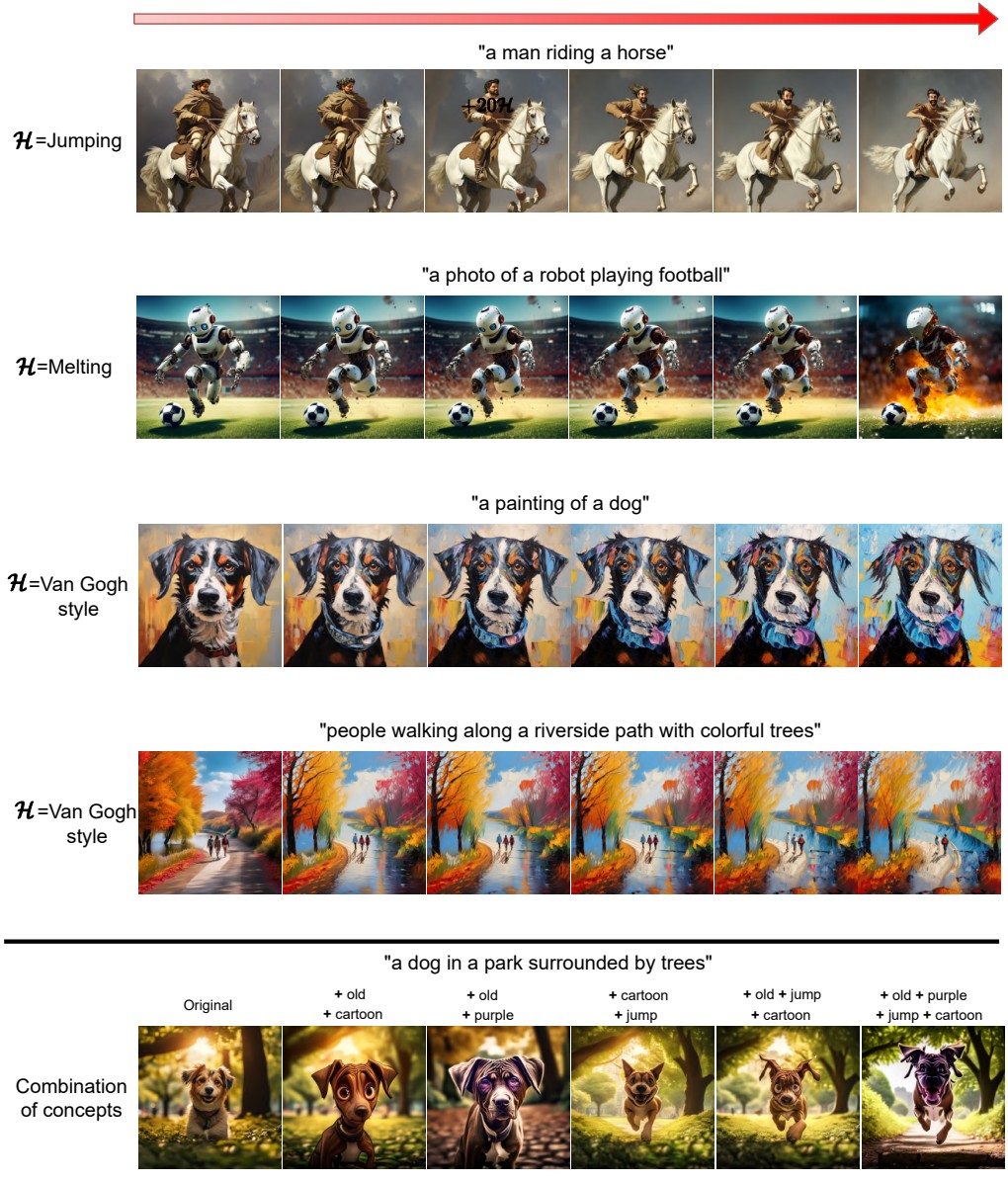

*Figure 12.* Concept addition on non–human-centric prompts. We apply learned concept vectors for style, color, motion (e.g., jump), and object state to non–human-centric text prompts for both scaling and linear composition applications. The examples demonstrate that the same mechanism generalizes beyond human-focused prompts and enables direct semantic control through the linear structure of the learned concept vectors.

| Prompt | Original | + Red |
|---|---|---|
| "Cinematic photo of a modern car on a rain-soaked street at dusk, neon reflections on wet asphalt, urban haze, volumetric light, glossy bodywork, ultra-realistic, moody atmosphere." | | |

| | Original | + Cartoon style |
|---|---|---|
| "A detailed illustration of a panda in a zoo setting, surrounded by bamboo, rocks, and soft greenery, with bold outlines, cel-shaded surfaces, expressive eyes, slightly exaggerated proportions." | | |

| | Original | + Male |
|---|---|---|
| "A nurse with a calm expression, standing at the nurse station, surrounded by computers and medical charts, hospital corridor in the background, cinematic lighting." | | |

| | Original | + Young |
|---|---|---|
| "A dog wearing sunglasses and a Hawaiian shirt relaxing on a beach chair next to a surfboard." | | |

*Figure 13.* Generations under complex compositional prompts (with the last two prompts identical to those in (He et al., 2025)). The examples illustrate that our framework reliably injects the desired concepts while maintaining strong fidelity to the full input prompt, even for intricate and highly compositional descriptions.

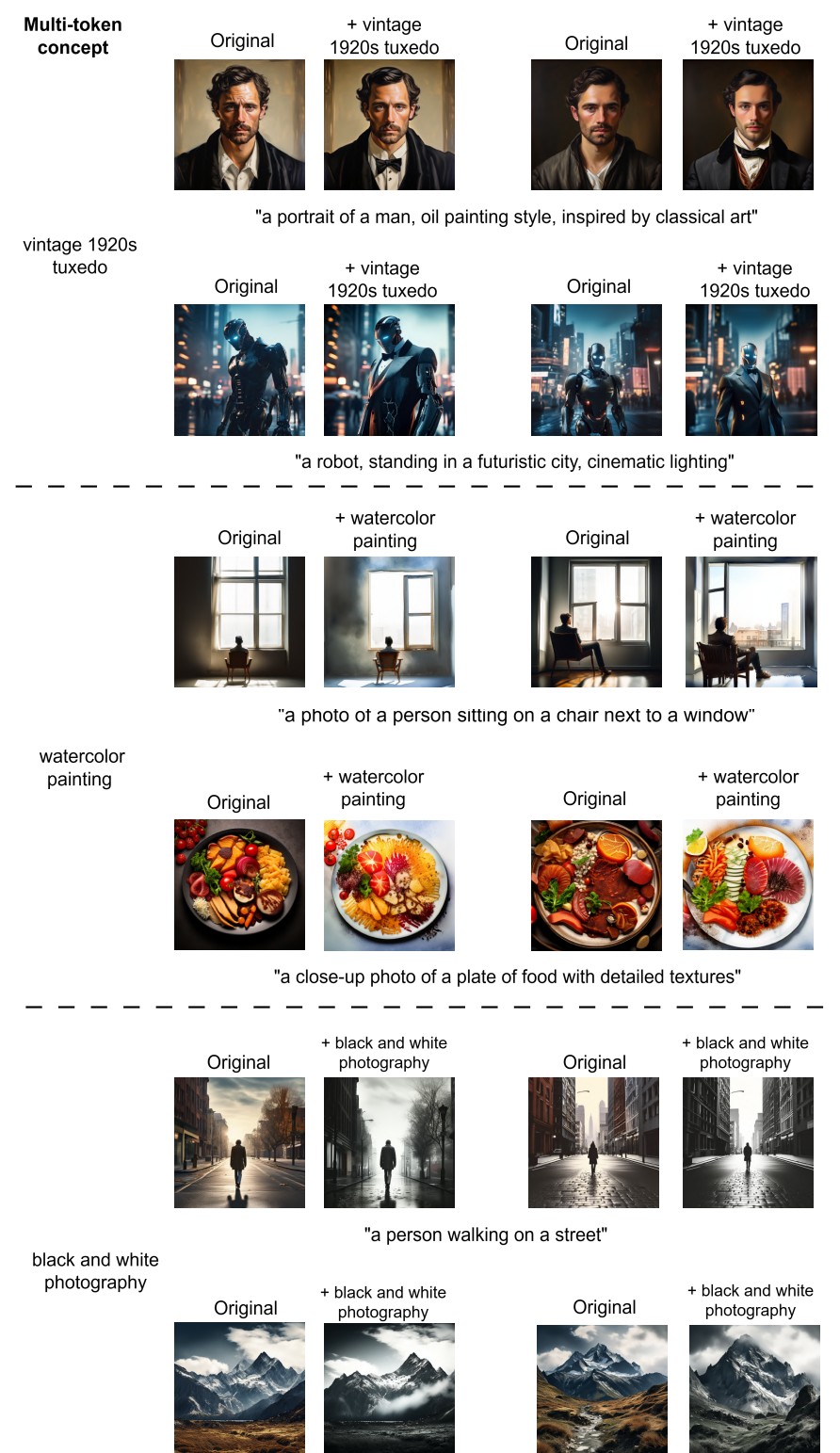

*Figure 14.* Qualitative results for multi-token concept injection. We learn concept vectors for four compositional attributes: "vintage 1920s tuxedo", "watercolor painting", "hospital room", and "black and white photography", and apply them across diverse input prompts. Our method consistently achieves the intended semantic transformation while preserving scene layout, subject identity, and background fidelity, demonstrating robust handling of complex, multi-token concepts.

