# OpenReview forum: "Responsible Text-to-Image Diffusion: Interpretable and Linearly Controllable Semantics for Fair and Safe Generation"
_ICML.cc/2026/Conference — ICML 2026 regular_

### Official Review · Reviewer_fDDu · 2026-02-23

**Soundness:** 3
**Presentation:** 2
**Significance:** 1
**Originality:** 2
**Overall Recommendation:** 3
**Confidence:** 3

**Summary:**

The challenge addressed by this article is the persistent bias and safety issues in T2I diffusion models, particularly for DiT based architectures. To solve this, the authors propose an architecture-agnostic framework that learns external, interpretable, and linearly controllable concept vectors. By systematically identifying the "Linear Semantic Locus" (LSL), the authors inject these learned vectors to steer generation away from stereotypical or unsafe outputs without fine-tuning the base model parameters. The method achieves state-of-the-art (SOTA) performance on bias mitigation benchmarks (e.g., WinoBias) across recent models like FLUX, PixArt, SD3.5, and SDXL.

**Compliance With Llm Reviewing Policy:**

Affirmed.

**Final Justification:**

4 weak accept

**Key Questions For Authors:**

1. Can you provide a cost-benefit analysis or an empirical comparison against an LLM-based prompt-rewriting baseline? How does the computational overhead of training specific concept vectors justify the performance gains over simply appending demographic keywords (e.g., "male", "African American") to the input prompt?
2. Could you provide aesthetic quality scores (e.g., using an aesthetic predictor) or human-evaluation studies to explicitly prove that the addition of the concept vector into the attention heads does not degrade the aesthetic value of the generated images?
3. Are there any quantitative evaluations planned for the broader applications shown in Appendix F (e.g., using CLIP directional metrics for style changes or object state modifications)?

**Limitations:**

While the authors include a dedicated "Impact Statements" section (Section 6) that adequately discusses the potential dual-use risks and negative societal impacts of their work (e.g., the potential misuse of the framework to amplify harmful stereotypes rather than mitigate them), they do not adequately discuss the technical limitations of their proposed framework.

**Constructive suggestions for improvement:** The authors should add a dedicated paragraph discussing technical limitations, specifically addressing: 1) the computational overhead required to gather data and optimize a new concept vector for every single desired attribute; 2) potential failure cases where the near-linearity assumption might break down (e.g., highly entangled, rare, or overly abstract concepts); and 3) how the method scales or behaves with extremely long or complex prompts where attention weights across the ViT heads are highly dispersed.

**Strengths And Weaknesses:**

**Paper Strengths**
1. **Innovative Application to ViT-based Diffusion Models:** The paper is pioneering in its exploration of bias mitigation specifically tailored for DiT-based T2I models. Identifying that multi-modal attention heads encode interpretable and near-linear semantic structures. Overall, an important concept analyzed by this paper is how specific attention heads can be manipulated in a linear fashion to control high-level semantics.
2. **Strong Empirical Results on Fairness:** The proposed method achieves state-of-the-art results on standard benchmarks. Compared to baselines like EMBEDIT and Weak Guidance (WG), the proposed concept-alignment loss effectively minimizes the deviation ratio for both gender and race across multiple cutting-edge models, demonstrating robust bias reduction.

**Paper Weaknesses**
1. **Marginal Contribution over Simple Prompt Engineering & Potential Quality Degradation:** The practical necessity and problem definition of the proposed method are somewhat weak. In real-world applications, mitigating demographic bias (gender, race) or filtering unsafe content can often be achieved through straightforward, training-free prompt engineering, such as using an LLM to rewrite a brief prompt like "a photo of a baker" to explicitly include "male" or "female". Training concept vectors for thousands of steps is computationally expensive and complex compared to these simple textual interventions. Furthermore, while the authors provide FID and CLIP scores to argue that image quality is preserved, injecting vectors inevitably perturbs the pre-trained latent knowledge. The evaluation lacks explicit aesthetic quality metrics (e.g., LAION aesthetic scores or human preference studies) to guarantee that the visual appeal of the generated images is not implicitly degraded during this interference.
2. **Lack of Quantitative Evaluation for Broader Applications:** The rigorous experimental scope is largely confined to human-centric attributes (gender, race) and safety. While the authors claim the method generalizes to other domains and provide qualitative examples in Section 4.2 and Appendix F (e.g., style transfer, object melting, motion), there is a lack of quantitative evaluation for these non-human-centric edits. Since theoretical linear vector manipulation should easily extend to broader applications like image morphing and compositional generation, the absence of quantitative benchmarks for these tasks leaves the true scalability and generalizability of the proposed method unproven.

---

> ### Author Rebuttal · Authors · 2026-03-27
>
> **Q1** We thank the reviewer for all constructive suggestions. We agree that a direct comparison against an LLM-based prompt-rewriting baseline would strengthen the paper, and we **will include such a comparison in the revision, along with a clearer cost–benefit analysis**. Also please refer to Response Q3 (Reviewer TEJ2) for a detailed computational analysis.
>
> **Cost–Benefit Trade-off**: The key trade-off is between **zero-training, per-request cost** (prompt rewriting) and **one-time training with negligible inference cost** (ours). Prompt rewriting requires an additional LLM forward pass for every query, introducing latency, GPU overhead, and potential stochasticity, which is critical in real-world deployments.  In contrast, our method incurs a one-time offline training cost, after which inference reduces to a simple vector addition in a few internal layers, with effectively zero overhead. While prompt rewriting is cheaper for one-off usage, our approach is more efficient in repeated or large-scale deployments, where eliminating per-prompt LLM calls significantly reduces latency and operational cost. Here are key reasons/results showing **our method provides substantially stronger improvements beyond prompt engineering**:
>
> ### Why Training is Necessary (vs. Keyword Appending)
> 1. **Direct correction of internal bias:** Prompt editing cannot fully address biases embedded in the denoiser. Our method intervenes directly in the generative process.
> 2. **Semantic disentanglement:** Keywords often introduce entangled attributes (e.g., gender affecting age). Our concept-alignment loss explicitly suppresses such leakage.
> 3. **Linear controllability and composability:** Learned vectors can be scaled and combined (e.g., gender + age + style), enabling fine-grained control that is difficult to achieve reliably with prompts.
> 4. **Deployment Motivation:** Importantly, our method addresses a realistic setting where prompt modification is not permissible:
>
> - **User intent preservation:** rewriting prompts may alter meaning and reduce trust.
> - **System constraints:** prompts may originate from upstream systems where modification is not allowed.
> - **Compliance and auditability:** original inputs must remain unchanged.
> - **Robustness:** LLM rewriting can introduce variability and semantic drift, especially for complex prompts.
>
> 5. **Empirical Evidence:** Although we do not yet include an explicit LLM rewriting baseline, the paper already compares against strong text-side methods (WG and EMBEDIT). Across PixArt, FLUX, and SD3.5, our method consistently achieves substantially lower deviation ratios while maintaining or improving CLIP alignment.
>
> Overall, we do not claim that concept vectors universally replace prompt rewriting. Rather, our method directly addresses bias within the denoising model, providing improved fairness, stability, controllability, and zero per-request cost in deployment. We will clarify this positioning and include direct prompt-rewriting comparisons in the revision.
>
> **Q2** To address this, we computed aesthetic scores (AS) using the LAION aesthetic predictor (ViT-L/14) across all 36 occupations for gender societal group (100 images each). As shown in Table 1, our method achieves aesthetic scores nearly identical to the original unmodified DMs across all pre-trained backbones (e.g., 6.6864 vs. 6.6109 for FLUX). This confirms that concept vector addition does not degrade the perceptual or aesthetic quality of generated images, complementing our FID and CLIP results.
>
> Table 1: AS comparison between the original DM  and our method across PixArt, FLUX, SD3.5, and SDXL.
> | Model         | PixArt | FLUX   | SD3.5 | SDXL  |
> |--------------|--------|--------|-------|-------|
> | Original DM  | 6.9674 | 6.6864 | 5.9984 | 6.4946 |
> | Ours         | 6.8649 | 6.6109 | 5.8780 | 6.4197 |
> | Difference   | 0.1025 | 0.0755 | 0.1204 | 0.0749 |
>
> **Q3** To address this, we now provide quantitative evaluation for the broader applications presented in Appendix F. Specifically, we report **CLIP score and AS** for 100 generated images per prompt in Table 2.  These results show that our method maintains competitive image quality (AS) while enabling controlled semantic transformations and preserves or improves CLIP alignment compared to the original models.
>
> Table 2: Quantitative evaluation of broader concept control across different prompts and concepts.
> | Prompt | Concept | CLIP↑ (Original/Ours) | AS↑ (Original/Ours) |
> |-------|--------|-------------------|------------------|
> | A detailed illustration of a panda in a zoo setting... | Cartoon | 34.03 / 34.00 | 7.11 /7.10 |
> | Cinematic photo of a modern car on a rain-soaked street at dusk... | Color | 32.56 / 32.51 | 7.03 / 7.02 |
> | People walking along a riverside path with colorful trees | Van Gogh | 32.34 / 32.30 | 7.78 / 7.76 |
> | A photo of a robot playing football | Melting | 28.54 / 28.51 | 6.65 / 6.59 |

---

> > ### Author Rebuttal · Reviewer_fDDu · 2026-04-03
> >
> > I have reviewed the authors' rebuttal and the additional experimental data provided. The inclusion of Aesthetic Scores (AS) and the quantitative evaluation of broader applications (Table 1 and 2 in the rebuttal) effectively address my primary concerns regarding potential image quality degradation and the lack of generalization assessment. The authors' clarification on the cost-benefit analysis compared to prompt-rewriting is convincing from a system deployment perspective. My concerns have been resolved. I will adjust my score to reflect the improved empirical rigor of the paper.

---

### Official Review · Reviewer_TEJ2 · 2026-03-03

**Soundness:** 2
**Presentation:** 3
**Significance:** 2
**Originality:** 2
**Overall Recommendation:** 4
**Confidence:** 4

**Summary:**

This article focuses on the mitigation of social biases and the enforcement of safety in modern Text-to-Image (T2I) diffusion models, specifically bridging the gap between U-Net architectures and ViT-based models. The authors propose an architecture-agnostic approach called "Linear Semantic Locus" (LSL). For ViTs, the authors identify the multi-modal attention heads as the equivalent LSL, demonstrating that these heads encode interpretable and near-linear semantic structures. This loss distills the semantic difference between a target-included prompt and a target-removed prompt into a vector that can be added to the LSL of a frozen model. And the numerical results show that this method effectively mitigate the social biases and safety for various kinds of generative models.

**Compliance With Llm Reviewing Policy:**

Affirmed.

**Final Justification:**

In summary, most of my concerns have been adequately addressed during the rebuttal. My final justification is neutral, but leaning positive.

**Key Questions For Authors:**

1. **Critical Theoretical Inconsistency about Flow Matching vs. DDPM**: This is a major issue. The paper consistently formulates the problem using standard DDPM notation ($\epsilon$-prediction) in Section 3 and provides a standard DDPM sampling algorithm (Algorithm 3) in Appendix. However, FLUX and SD3.X typically use Flow Matching training objectives ($v$-prediction), rather than the noise $\epsilon$ directly. Typically, Algorithm 3 Line 996 also uses the standard DDPM sampling step: $x_{t-1} \leftarrow \frac{1}{\sqrt{\alpha_t}} (x_t - \dots \hat{\epsilon}_t)$. This sampling schedule is mathematically incorrect for Flow Matching models, which require ODE solvers like Euler steps. Loss function Eq.1 minimizes the $L_2$ norm of $\epsilon$ differences. If applied to a Flow Matching model predicting $v$, the semantic scaling might behave differently than in $\epsilon$-space. Despite the main differences between FM and DDPM training objectives are the schedule, the authors fail to clarify whether they converted the Flow Matching output to $\epsilon$-space. If they applied the standard DDPM sampling code (Algorithm 3) to a FLUX checkpoint, the results should be insufficient, raising questions about the implementation details. This mismatch needs to be clarified during the rebuttal period.

2. **Definition of LSL**: The LSL in ViT Multi-Modal Attention heads) is defined somewhat heuristically. The authors claim these heads are the LSL, but it appears they selected the components that worked best. The theoretical justification for why attention heads must exhibit the same linearity as a compressed bottleneck in U-Net is weak; it is presented more as an empirical observation than a derived property. Also, in Section 4 (Layer Selection), the authors mention ablating layers to find the most important ones (e.g., layers 3-18 for FLUX). However, this selection seems to be a hyperparameter tuned per model.

3. **Computational Overhead in Training**: The method requires three copies of the full T2I model in memory during the training phase (Model $\mathcal{M}_1, \mathcal{M}_2, \mathcal{M}_3$ in Figure 2). For massive models like FLUX (12B), keeping three copies plus gradients for the vector optimization is inapplicable. The paper mentions using an H100 (80GB), but this high resource requirement limits accessibility compared to methods that only require inference-time adaptation or embedding optimization.

4. **Limited Extensibility of Concept Vectors**: The generalization capability of the proposed concept vectors is insufficiently explored. First, the experiments focus on concrete attributes (e.g., gender, style), leaving the method's effectiveness on abstract concepts (e.g., "professionalism" or "cultural neutrality") unverified. Second, the reliance on closed datasets (e.g., WinoBias) raises concerns about cross-scenario adaptation; it is unclear if vectors trained on specific professions remain effective for unseen domains without retraining.

**Limitations:**

As previously noted, a primary concern is the discrepancy between the proposed methodology and the actual implementation (specifically regarding the distinction between FM and DDPM). This inconsistency undermines the theoretical soundness of the paper. Furthermore, while the contributions are primarily empirical, the current numerical results are insufficient to substantiate the central claims.

**Strengths And Weaknesses:**

**Stengths**

* As the field shifts towards DiT (Diffusion Transformer) architectures like FLUX and SD3, existing safety and editing tools designed for Stable Diffusion 1.5/XL (U-Net) are becoming obsolete. The proposed framework is unified and architecture-agnostic.

* The Design of concept-alignment loss is a non-trivial approach. By freezing the model and optimizing only the additive vector to match the noise prediction of a prompt containing the concept, the authors achieve high disentanglement. The numerical results show that this objective minimizes correlation better than simple reconstruction losses.

* The analysis in Section 3.2 and Appendix B, particularly the PCA analysis of attention heads, provides detailed insight into how semantic attributes are distributed in ViT-based diffusion models. Identifying that specific heads act as "linear controls" is a useful finding for the community.

**Weaknesses**
Please see the questions below.

---

> ### Author Rebuttal · Authors · 2026-03-30
>
> **Q1** We thank the reviewer for all careful suggestions. We agree that clarification is needed.
> **On notation (Algorithm 3 and Eq. 1):**  FLUX and SD3.5 use flow matching with velocity prediction, while Algorithm 3 is written in DDPM-style notation for simplicity. In our implementation, we use the **model-native schedulers from the diffusers library**, which correctly apply ODE-based solvers (e.g., Euler) for flow-matching models. Algorithm 3 should therefore be interpreted as a schematic description rather than a literal sampling procedure. We will clarify this in the revision with proper math notations.
>
> **On the concept-alignment loss:**  In the loss terms are computed from the same frozen model at the same timestep and latent, so they lie in the same output space by construction. As a result, the loss depends only on **relative differences between conditioned outputs**, and is therefore **invariant to whether the model uses $\epsilon$-prediction or $v$-prediction**.
>
> **On where $\mathcal{H}$ operates:**  The external concept vector $\mathcal{H}$ is added at the LSL without affecting any internal parameter of the model. This is upstream of the output parameterization and therefore independent of the sampling formulation.
>
> **Q2** We clarify that LSL identification is not heuristic tuning, but follows a unified, evidence-driven procedure. First, LSL is defined as regions with approximately linear semantic structure, supported by PCA showing dominant components aligned with interpretable attributes, similar to U-Net bottlenecks. Second, quantitative results (W6, reviewer G4gB) show strong linearity (e.g., $R^2 = 0.955$ for motion, $R^2 = 0.869$ for style), indicating predictable semantic control not uniform across layers. Third, mean-ablation provides causal evidence: ablating non-critical early layers has little effect for all models. Please also see W1 and W2, reviewer G4gB.
>
> **Q3**
> Regarding memory, Fig. 2 is conceptual and does not require loading three separate DMs. As shown in Appendix D, $M_1$ is used only in preprocessing, while the remaining branches reuse the same frozen model, enabling sequential forward passes with memory comparable to standard diffusion training.  We report runtime per epoch for PixArt, FLUX, and SDXL (Table 1, H100 GPU). Although the method introduces a one-time concept learning stage, it enables interpretable, near-linear semantic control not supported by inference-time methods (please see Q1, reviewer fDDu). Inference remains efficient, with negligible overhead since concept injection involves only lightweight linear operations.
>
> | Category | Metric | Models |
> |----------|--------|--------|
> | Training Time| Runtime per epoch (3k images) | PixArt: 4.25 min, FLUX: 10.52 min, SDXL: 1.40 min |
> | Inference Time| Runtime per image (1024×1024) | PixArt: 7.14 sec, FLUX: 6.97 sec, SDXL: 2.36 sec |
> | Inference Time| Runtime per image with adding concept H(1024×1024) | PixArt: 7.32 sec, FLUX: 7.13 sec, SDXL: 2.48 sec |
>
> Finally, we analyze the effect of training length by evaluating gender fairness using 150 generated samples for the prompt "a photo of a doctor". The quantitative results (Table 2) show that deviation ratio improves with training steps and stabilizes after 15k steps, indicating that effective concept learning does not require excessive training.
>
> | Metric | 1k Steps | 5k Steps | 10k Steps | 15k Steps | 20k Steps |
> |--------|----------|----------|------------|------------|-----------|
> | $\Delta$ | 0.09| 0.07 |0.06 | 0.05 | 0.05 |
>
>
> **Q4**  We agree that evaluating broader generalization is important, and provide additional quantitative results in Q3 (Reviewer fDDu), Q2 (Reviewer Lubo), and W6 (Reviewer G4gB), with further results in Appendix F. Across these experiments, the proposed concept vectors extend beyond concrete attributes and generalize to a broader class of visually grounded concepts, achieving strong performance even for rare/OOD attributes (e.g., $R^2 > 0.77$). Regarding cross-scenario generalization, the learned concept vectors are not tied to specific prompts or datasets, but instead capture **representation-level semantic directions**. In practice, we observe that vectors trained on one set of prompts generalize across diverse prompt formulations, including unseen human-centric and non-human-centric scenes. For example, we learn the “jump” concept using images of "a person jumping", and then successfully transfer this concept to non-human scenarios, as shown in Appendix E. This is consistent with our design, where the concept-alignment loss operates on **image-level differences rather than prompt-specific supervision**, enabling reuse across domains without retraining (Please also see W2, Reviewer LUbo). While WinoBias is used as a controlled benchmark for fairness evaluation, we further validate generalization through experiments on COCO-30k, I2P, and diverse prompt sets, demonstrating that the learned vectors transfer beyond the training distribution.

---

> > ### Author Rebuttal · Reviewer_TEJ2 · 2026-04-03
> >
> > I have read the authors' response, and the scores have been adjusted accordingly.

---

### Official Review · Reviewer_LUbo · 2026-03-10

**Soundness:** 1
**Presentation:** 2
**Significance:** 1
**Originality:** 2
**Overall Recommendation:** 3
**Confidence:** 4

**Summary:**

The paper proposes a method to learn interpretable and linearly controllable "concept vectors" for text-to-image diffusion models, specifically targeting ViT-based architectures.

**Compliance With Llm Reviewing Policy:**

Affirmed.

**Final Justification:**

Thanks for the authors' detailed rebuttal. I maintain my score.

**Key Questions For Authors:**

1.	Learning a unique vector for every concept (gender, race, style, etc.) is computationally cumbersome compared to unified prompt-tuning or multi-concept adapters.

2.	It remains unclear if this linear property holds for highly rare or out-of-distribution visual attributes.

3.	Since you claim "near" linearity, can you provide a quantitative measure of the non-linear distortion when scaling concept vectors to extreme values?

4.	How does the performance degrade if concept vectors are applied to non-optimal layers?

5.	In what scenarios does the "near-linear" assumption fail for ViT attention heads?

**Limitations:**

yes

**Strengths And Weaknesses:**

Strengths:

The paper is logically structured, and the concept-alignment pipeline is clearly illustrated in Figure 2 .

The paper discusses an important topic, finding architecture-agnostic ways to ensure "Responsible AI".

Weaknesses:

1.	The mechanism for identifying "most suitable layers" is primarily empirical and lacks a rigorous theoretical justification for why certain layers are more semantic than others.

2.	The "target-removed" prompt construction ($\Gamma=\Phi\backslash\mathcal{T}$) is overly simplistic; it assumes attributes can be cleanly subtracted from natural language, which often fails for entangled concepts.

3.	Results indicate performance plateaus at 3,000 images, suggesting the method might struggle to learn nuanced or long-tail attributes that require more diverse data .

4.	The concept of adding steering vectors to latent spaces is well-established; the novelty lies primarily in the location (attention heads).

5.	The term "architecture-agnostic" is slightly hyperbolic as it still requires manual identification of specific architectural "loci" for each new backbone.

---

> ### Author Rebuttal · Authors · 2026-03-29
>
> **(W1)**  We thank reviewer for all constructive feedback. Layer selection is empirically motivated but supported by representation evidence and ablation. We define the Linear Semantic Locus (LSL) as regions with near-linear, disentangled semantics. In U-Net, this corresponds to the bottleneck. In ViT-based models, our PCA analysis over attention heads shows principal components align with interpretable attributes (e.g., gender), indicating near-linear structure. Mean-ablation shows early layers have minimal effect, while semantic collapse occurs in mid layers. Thus, selection is grounded in linear structure (PCA) and causal importance (ablation), not heuristic. Please refer to W1 (Reviewer G4gB) for more details on adding learned concept vectors at different points.
>
> **(W2)**  We respectfully clarify that our **method does not rely on the assumption that attributes can be cleanly removed at the linguistic level**. Importantly, the concept vector is learned to capture the **visual attribute direction in the model’s representation space**, driven by image-level supervision rather than textual correctness (i.e., the loss is computed from image-level differences rather than prompt-level supervision). The “target-removed” prompt serves only as a weak conditioning scaffold, while the optimization enforces consistency between model predictions across paired inputs. As a result, the learned concept vector encodes consistent visual variation across samples, rather than depending on precise prompt subtraction. This makes the method robust to imperfect prompts. In practice, concept vectors generalize across diverse prompt formulations, including human-centric and non-human-centric scenes (please see Q3 reviewer fDDu), indicating representation-level semantics rather than prompt artifacts.
>
> **(W4)**  While vector-based steering has been explored in prior work, our contribution goes beyond merely selecting an injection location. First, the proposed **concept-alignment loss** is non-trivial: it learns a concept vector by enforcing consistency between diffusion trajectories with and without the target attribute, enabling representation-level semantic directions. This differs fundamentally from prior approaches that rely on simpler loss formulations (please see Appendix A). Second, we introduce a layer-wise mean-ablation study to analyze the role of MCA modules across layers. This provides a causal understanding of where semantic information resides. Third, we identify locations exhibiting near-linear properties in ViT-based diffusion models. Together, these contributions establish a principled framework for learning and localizing semantic directions, rather than simply selecting an injection point.
>
> **(W5)**  We used the term *architecture-agnostic* rather than *model-agnostic* to reflect this distinction. Our method does not require architecture-specific modifications or retraining, but it does involve identifying loci within each backbone and we will clarify this wording in the revision.
>
> **(Q1)**  Please refer to Q1 (Reviewer fDDu) for reasons/results showing our method provides substantially stronger improvements beyond prompt-tuning or adapters.
>
> **(Q2)**  We agree that evaluating rare or OOD attributes is important. To address this, we extended our scaling-linearity analysis to visually less frequent concepts such as "origami texture" and "bioluminescent effects". Using the same protocol as in W6 of reviewer G4gB, we measure linearity via the coefficient of determination ($R^2$) between semantic strength and intervention magnitude. As shown in Table 1, near-linear behavior is preserved for OOD attributes, with $R^2 = 0.802$ (origami) and $R^2 = 0.775$ (bioluminescence), compared to $R^2 = 0.911$ for a common attribute. This indicates that semantic modulation remains predictable and monotonic, although slightly weaker for less frequent concepts. Additional results will be included in Appendix F.
>
> **Table 1. Linearity for common and OOD attributes**
> | Attribute Category | Concept | $R^2$ |
> |---|---|---:|
> | Common | Gender (Female) | 0.911 |
> | Rare / OOD | Origami Texture | 0.802 |
> | Rare / OOD | Bioluminescence | 0.775 |
>
> **(Q3)**  Please refer to Response W6 (Reviewer G4gB).
>
> **(Q4)**  From the ablation study, removing critical layers leads to rapid degradation in semantic alignment, as reflected by significant drops in CLIP score and we provided effect of adding concept vectors beyond the identified optimal layers (Table 6, Appendix).  Extending interventions to all heads/layers yields no fairness gain and reduces CLIP (e.g., PixArt). Applying vectors to MLP instead of MCA is ineffective: with “a photo of a doctor,” results remain biased (95 male / 5 female), confirming localization is necessary.
>
> **(Q5)**  Linearity degrades in: (i) **very large magnitudes** (e.g., $|s|>40$) due to saturation, (ii) **highly entangled attributes** (e.g., male+mustache), (iii) **adding outside LSL**, where near-linearity drops.

---

> > ### Author Rebuttal · Reviewer_LUbo · 2026-04-05
> >
> > I thank the authors for their detailed response. After reviewing the rebuttal and the other reviewers' comments, I remain concerned regarding the following points:
> > 1. The layer selection process lacks a principled foundation, appearing largely heuristic without a clear explanation of the underlying architectural behavior.
> > 2. The "target-removed" prompt construction ($\Gamma = \Phi \backslash \mathcal{T}$) remains abstract. It is unclear how this subtraction logic scales to complex, multi-token, or entangled concepts where semantic boundaries are not cleanly partitioned.
> > 3. The reliance on aggregate metrics (e.g., averages across 36 professions) may mask localized failures. Furthermore, the submission lacks a deep ablation study demonstrating how the layer identification mechanism specifically governs the trade-off between attribute controllability and image fidelity across varying backbones.
> > 4. The early performance plateau (at 3k images) suggests the method may lack the capacity to model the complexity of rare or highly specific visual concepts.

---

> > > ### Author Response · Authors · 2026-04-06
> > >
> > > **1.** Thank you for your follow-up questions and constructive feedback. We emphasize that layer selection is grounded in two complementary signals rather than a heuristic choice. PCA identifies attention heads with near-linear semantic variation, where principal components align with interpretable attributes such as gender and pose (Fig. 3/9). Mean-ablation establishes causal necessity: layers whose removal causes semantic collapse are critical, while early layers with minimal impact are excluded. This aligns with established mechanistic interpretability practices (Ahn et al., 2025; Gandelsman et al., 2024 {in manuscript}).
> > >
> > > New results provided in following Table 1, together with Table 6 (Appendix) validate this: restricting to Top-5 heads yields the best CLIP while matching or improving deviation ratio, whereas removing key heads degrades performance, confirming the loci are necessary and sufficient.
> > >
> > > In the revision, we will further provide: (i) layer-wise linearity via PCA (explained variance and direction consistency), (ii) correlation with controllability and fidelity, and (iii) extended mean-ablation showing semantic degradation vs. cumulative layer removal. Together, these establish LSL as regions with both linear structure and causal semantic influence, providing a principled foundation beyond empirical observation.
> > >
> > > **Table 1**
> > > | Heads | Δ ↓| CLIP ↑|
> > > |----------------|----|------|
> > > | All heads| 0.01 | 31.62|
> > > | Top-5 heads| 0.01 | 34.01|
> > > | All except Top-5 | 0.02 | 30.26|
> > >
> > > **2.** We clarify that the target-removed prompt $\Gamma$ serves only as a weak contrastive conditioning scaffold and does not perform semantic decomposition. The concept vector is learned through image-level supervision: the concept-alignment loss enforces consistency between diffusion trajectories conditioned on $(\Gamma + H)$ and $\Phi$, capturing visual rather than linguistic differences. Thus, prompt subtraction need not be precise, as optimization is driven by internal representation changes. This explains why learned vectors generalize across neutral prompts such as "a person", "a portrait of a person", "a human subject", and remain effective across diverse prompts, including compositional prompts (Fig. 13). For highly entangled concepts (e.g., male + mustache), some interference is expected; we will explicitly include a limitations section in the revised manuscript to clarify these boundaries, supported by visual examples.
> > >
> > > To demonstrate multi-token capability, we provide qualitative results for “vintage 1920s tuxedo” via an anonymous link below. Due to time shortage, we limit this to visual results; the revision will include more quantitative and qualitative validation, including imperfect removals.
> > >
> > > visual results for multi-token concept:
> > > https://i.postimg.cc/5tTcY4rv/multi-tokens.png
> > >
> > > **3.** On aggregate metrics: Per-occupation results are reported in Tables 7 and 8 across 36 professions and four backbones, where localized variations are visible. For example, "constructor" and "hairdresser" show higher residual deviation in SD3.5, reflecting stronger visual stereotypes. We will also report per-profession variance and worst-case metrics in the revision.
> > >
> > > On the controllability–fidelity trade-off: We quantify this across PixArt, FLUX, and SD3.5 under three settings (Table 1 above and Table 6, Appendix). Top-5 head selection achieves the best CLIP while maintaining deviation ratio, whereas extending to all layers degrades alignment without fairness gains. Moreover, adding concepts into MLP yields 95 male / 5 female, confirming negligible controllability.
> > >
> > > In the revision, we will evaluate three regimes: (i) early layers, (ii) identified LSL layers, and (iii) non-LSL modules, which will provide more consistent structure.
> > >
> > > **4.** We respectfully reframe this concern: the plateau at ~3k images reflects semantic saturation rather than a capacity limitation. As our focus is bias and fairness, Fig. 10 reports deviation ratio, and target concepts (e.g., gender, race) are relatively common. Once the visual distribution is sufficiently covered, additional samples become redundant and add little new information, consistent with standard representation learning.
> > >
> > > Importantly, this does not imply limited capacity for complex or rare concepts. We report $R^2$ values for a common concept ("female") and a more complex OOD concept ("origami texture"), trained with 3k and 3.5k images (Table 2 below). The linearity of the OOD concept further improves with additional data, indicating that the method can effectively leverage more samples when needed. Overall, the plateau reflects efficient convergence for simpler concepts while retaining capacity for more complex attributes.
> > >
> > > **Table 2**
> > > | Concept Category | Concept| 3k Images ($R^2 \uparrow$)| 3.5k Images ($R^2 \uparrow$)|
> > > |----------------|---------------|--------------------------|----------------------------|
> > > | Common| Female| 0.91| 0.93|
> > > | OOD | Origami Texture | 0.80| 0.85|

---

### Official Review · Reviewer_G4gB · 2026-03-10

**Soundness:** 2
**Presentation:** 3
**Significance:** 2
**Originality:** 2
**Overall Recommendation:** 4
**Confidence:** 4

**Summary:**

This paper introduces "concept vectors" for modern ViT-based text-to-image diffusion models. Concept vectors allow steerable generation using interpretable and "linearly separable" vectors added to frozen diffusion model backbones. The key insight is that the attention heads in diffusion ViTs encode structured, disentangled semantic information analogous to the bottleneck layer in U-Net models. The authors train these external concept vectors using a "concept alignment loss", enabling conditional generation by isolating target attributes like gender, race, style. Generation via these concept vectors preserves image quality while enabling more equitable generation of concepts.

**Compliance With Llm Reviewing Policy:**

Affirmed.

**Final Justification:**

**POST REBUTTAL:** I thank the authors for their measured response. I will increase my score, contingent on them incorporating all discussed material as well as pending experiments into the final revision. Thanks for the discussion.

The strengths of this paper for me now outweigh the weaknesses given the discussion during the rebuttal, and I think the authors' empirical contributions are valuable to share with the broader community.

**Key Questions For Authors:**

Most of my questions are in the fields above. A few more are:

**(Q1)**: The deviation ratio metric treats fairness as uniform distribution across demographic groups. Is the uniform distribution always the right target? The reliance on classifiers to determine gender and race categories introduces its own biases, which are not discussed.

**(Q2)**: What does the concept-vector steered output's FID look like when compared with real ground truth images? In Table 1, FID is computed against images generated by the original pretrained model and not against real images. This is measuring distributional shift from the original model, not absolute image quality.

**(Q3)**: What happens when two concept vectors that are not semantically orthogonal are composed (eg: "young" and "female," which may be correlated in the training data)? Do the linear composition properties still hold?

**(Q4)**: The head importance mechanism (eq 2) selects the top-K heads. How sensitive is performance to K?

**Limitations:**

Mostly discussed, but I do have a worry that this technique is equally applicable to create "dangerous" concept vectors and could be used to augment models to produce illicit or harmful content. Some of this is discussed in the impact statement, but a more thorough discussion would be valuable, including potential safeguards.

**Strengths And Weaknesses:**

**Strengths**:
**(S1)**: The paper is well presented and is very clear. The problem is framed well and the paper appropriately frames its contributions.

**(S2)**: The paper appears to be one of the first to tackle concept-vector based generation for modern ViT-based conditional diffusion models, while still maintaining compatibility with UNet-style stable diffusion models.

**(S3)**: The method appears to work in steering generation leading to reduced inequity, especially for socially sensitive concepts e.g. associating CEOs predominantly with a particular gender and race.

Overall, the paper is clear and well-framed. The method is simple and effective.

**Weaknesses**:
**(W1)**: Limited originality. It is not that surprising that the multi-modal attention heads encode semantically separable concepts to do with text-to-image generation, especially given that these layers are where image and text information is fused. More analysis could be done here - do specific projections within attention layers affect generation more? E.g. do query vectors influence steerability more or less? or key or value vectors. The authors can do more analysis to identify precise contributions to steerability.

**(W2)**: Ablation - similar to the last point, more can be done on the ablation study. What happens if we add concept vectors to the MLP layers? If we add concept vectors to the timestep conditioning vector? How sensitive is the ablation method for determining "important layers" to prompt distributions? It seems that the layers chosen for concept vectors can differ quite widely across models. I have concerns about the method's robustness that might be addressed via more extensive ablations.

**(W3)**: Efficiency - this is a training method, and each concept vector needs to be independently trained. What is the computational cost of training each concept vector? How does length of training affect concept vector efficacy? Would one need to keep training additional concept vectors for each new concept, for each new model? What are the memory requirements for running 3 models simultaneously during training. It seems potentially wasteful and tedious. How is inference efficiency affected by concept vectors vs without?

**(W4)**: Baselines - given that this is a training method, there are multiple other ways to steer conditional generation that are also viable candidates for comparison. For example, PEFT techniques like LoRA or ControlNets, where each LoRA adapter could be merged in and out of the weights and trained using the same concept loss. The tradeoffs of alternative approaches are not sufficiently discussed. And this is just one example - there are other such methods in the literature that would make viable baselines.

**(W5)**: concept alignment - The alignment loss requires some degree of prompt engineering. The method also assumes the target attribute can be cleanly removed from a prompt, which may not be true (or may be non-trivial) for complex or illicit attributes. How does prompt phrasing affect the authors' proposed method?

**(W6)**: linearity - the linearity claims lack rigorous quantification. The paper claims near linear properties. But the evaluations are only done qualitatively through visual examples of scaling and composition (fig 1b, fig 12). There is no quantitative measure of linearity (e.g., deviation from perfect linear interpolation, or how well additivity holds across concept combinations).

Overall, the method seems simple and effective, but many questions remain as to the robustness and practicality of the approach. This leads me towards rejection.

---

> ### Author Rebuttal · Authors · 2026-03-26
>
> **(W1)** We thank the reviewer for all valuable suggestions. Our claim is more specific: these heads form a *practically usable linear semantic locus* enabling interpretable and linearly controllable concept vectors. To further support near-linearity, we conducted additional quantitative analysis in W6.
>
> **Projection-level ablation.**
> We perform a controlled ablation within MCA layers by injecting learned concept vectors into Q, K, and V, instead of the attention output (LSL). All other settings are fixed. Using 200 samples with the prompt "a photo of a doctor", results show:
>
> | Injection Point | Δ ↓ | CLIP ↑ |
> |-----------------|------|--------|
> | **LSL (Ours)** | **0.05** | **0.32** |
> | V | 0.12 | 0.27 |
> | Q | 0.24 | 0.24 |
> | K | 0.31 | 0.23 |
>
> LSL achieves the strongest controllability and alignment.
>
> **(W2)**
>
> **Intervention locus ablation.**
> Adding concept vectors into MLP blocks yields 95 male and 5 female outputs, indicating negligible controllability. This confirms semantic steering requires intervention at the multimodal fusion stage.
>
> **Layer robustness.**
> Empirical analysis (Sec. 3.3, Fig. 4) shows early layers contribute minimally. In PixArt, ignoring the first 4 vs 8 layers yields nearly identical results. Thus, layer selection mainly affects efficiency (rather performance) and is stable across architectures.
>
> **(W3)** Please refer to Response Q3 (Reviewer TEJ2).
>
> **(W4)** Our method differs fundamentally: the diffusion backbone remains frozen, and we optimize external concept vectors in a near-linear representation space. Unlike LoRA or ControlNet, which introduce nonlinear adaptations, our method relies on **simple additive** manipulation, enabling direct vector addition and scalar scaling for predictable compositional control. This linear composability is central and not the goal of existing PEFT approaches. Accordingly, we focus on baselines aligned with evaluating linear semantic or reporting their results based on most recent T2I DMs, though future comparisons are possible.
>
> **(W5)** Please note that our method does not require perfectly removing attributes from prompts. It relies on contrastive image-level supervision, where the concept vector captures **consistent visual variation rather than textual signals**. Neutral prompts act as weak scaffolds ( please see W4 in reviewer LUbo). This is why concept vectors generalize across diverse prompts. We validate robustness using multiple neutral prompts and evaluating on "a photo of a doctor". Results remain consistent (variance < 0.01), indicating robustness.
>
> **(W6)** We provide quantitative validation of near-linearity in the proposed LSL via scaling and additivity.
>
> **(1) Scaling linearity.**
> Let $Y$ denote the LSL representation and $H$ the concept vector. We apply:
> $ \tilde{Y}(s) = Y + sH $, where $s \in \mathbb{R}$. Semantic strength is defined as $ S(s) = \text{CLIP}(I_s, \Phi_H(=\text{a female person})) $. Fitting $ S(s) = \beta s + \epsilon $ yields strong linearity (e.g., $R^2 = 0.955$ for "jump", $0.869$ for "cartoon"), indicating predictable control. Linearity holds for $|s| \leq 30$, with mild saturation beyond.
>
> **(2) Additivity.**
> We measure intrinsic response: $ \delta_{\text{own}}(H) = S(Y + H) - S(Y) $. For $H_1, H_2$, the expected response is
> $ \delta_{\text{expected}} = \delta_{\text{own}}(H_1) + \delta_{\text{own}}(H_2) $, and the observed response: $ \delta_{\text{actual}} = S(Y + H_1 + H_2) - S(Y) $.
> Additivity is quantified as $ R_{\text{add}}^2 = 1 - \frac{\sum (\delta_{\text{actual}} - \delta_{\text{expected}})^2}{\sum (\delta_{\text{actual}} - \bar{\delta}_{\text{actual}})^2} $. We observe strong agreement:
> - young + jump: $R_{\text{add}}^2 = 0.914$
> - cartoon + van Gogh: $R_{\text{add}}^2 = 0.882$
> - young + female: $R_{\text{add}}^2 = 0.801$
>
> This confirms near-linear composability.
>
> **Q1**  Uniform is used as a standard baseline in prior fairness work. Our method is not restricted to this setting and supports flexible ratios (e.g., 25/75) via concept vector sampling. We acknowledge that evaluation tools may introduce biases, and will clarify that results reflect relative gains under a standard evaluation framework.
>
> **Q2**  We already report real-image FID on COCO to assess absolute realism (Table 2). In other cases, FID is computed against baseline outputs to measure distributional deviation, which we will mention in revision.
>
> **Q3**  Please refer to Response W6.
>
> **Q4**  The table below shows the gender societal group for "a photo of a doctor" using different scenarios for # of heads, where removing key heads has a larger impact than adding others. Please see Table 6 (Appendix) for additional comparisons.
> | Heads                |  Δ (↓) | CLIP (↑) |
> |----------------------|----------------------|----------|
> | All heads           | 0.01                   | 31.62       |
> | Top-5 heads         | 0.01                   | 34.01       |
> | All except Top-5    | 0.02                   | 30.26       |

---

> > ### Author Rebuttal · Reviewer_G4gB · 2026-04-03
> >
> > I thank the authors for their response. My concerns have been partially addressed.
> >
> > I do think, however, that LoRA and ControlNet are valid baselines. The backbone also remains frozen in both of these methods. Both methods also conceptually provide exactly the same kind of steerability as the method proposed in this paper.
> >
> > For runtime efficiency, thank you for providing the clarification that all 3 models are not loaded during training, and that for M2 and M3 the operations are sequential. However, the training times provided lack a reference - they need to be compared against equivalent baselines. Some details are also missing. How many epochs are used? What does 15k steps correspond to? Similar questions apply for inference - what is the difference in runtimes with vs without concept vectors, and then against baselines?
> >
> > I am still a bit skeptical of the response to W5. Yes, I understand that visual differences are captured, but this depends on visual differences in generated images with varying prompts cleanly correlating with semantic differences in the prompts themselves. This is true in simple prompts like "doctor" vs "female doctor" but much less clear for more complex prompts that chain together multiple attributes with more complex descriptions. As a simple example: "A person who is not smiling", how is negation dealt with? This is only one example, and there are many prompts that are more complex semantically that might not decompose in the manner the paper assumes.
> >
> > Also, minor point, I'm a bit unclear on the results of Table 2 in the response to reviewer fDDu, it seems that there are almost no differences compared to the original. Is this meant to show that the concept vectors are not regressing generation quality, or that they are enabling less biased generations?

---

> > > ### Author Response · Authors · 2026-04-05
> > >
> > > **1.**  Thank you for your follow-up questions and all constructive feedback. We agree that LoRA- and ControlNet-based approaches are relevant practical baselines, and we will include direct comparisons with [1] in the revised manuscript. Due to the limited rebuttal timeframe, we provide here a preliminary comparison using a LoRA-based method [1] and more detailed results will be provided in revised manuscript.
> > > Specifically, we conducted a small-scale experiment using method of [1] on the concepts female and male (500 training epochs) across two occupations (doctor and nurse) from the WinoBias benchmark. We report the average deviation ratio (Δ), CLIP, and AS (below Table) over 150 generated images per occupation. A key practical issue, also noted in [1], is that applying LoRA throughout inference can lead to instability and loss of identity. To mitigate this, the authors of [1] adopts an SDEdit-style schedule, activating LoRA only after a chosen timestep during denoising. This introduces an additional hyperparameter that must be manually tuned to balance concept strength and image quality.  Based on their proposed range we found timestep 700/1000 with best fair generation result. We report this preliminary comparison only on SDXL (as of now we were unable to obtain stable FLUX results using the released implementation).
> > >
> > > | Method                          | Δ ↓      | CLIP ↑    | AS ↑     |
> > > | ------------------------------- | -------- | --------- | -------- |
> > > | Original SDXL                   | 0.81     | 31.18     | 6.54     |
> > > | Concept sliders [1] | 0.17     | 30.14     | 6.12     |
> > > | Ours                       | 0.12| 31.15 | 6.44 |
> > >
> > > [1] Gandikota et al. "Concept sliders: Lora adaptors for precise control in diffusion models." In ECCV , 2024.
> > >
> > > **2.**  We apologize for any confusion caused by representing training and inference runtimes in a single table. We clarify and separate them below, including additional baselines:
> > >
> > > **Inference runtime**:
> > > In the table for reviewer TEJ2, the second row shows the original model runtime per image, while the third reports our method after adding the concept vector. We revise the presentation below for clarity and include additional baselines. All values are in seconds. As shown, the inference overhead of our method is negligible, as it involves only a simple addition operation.
> > > | Method      | PixArt   | FLUX     | SDXL     |
> > > | ----------- | -------- | -------- | -------- |
> > > | Original DM | 7.14| 6.97 | 2.36|
> > > | H-Guide     | N/A | N/A | 2.90|
> > > | Self-Dis    | N/A | N/A| 2.48|
> > > | WG          | 7.40| 7.21| 2.55|
> > > | **Ours**    | 7.32| 7.13| 2.48|
> > >
> > > **Training runtime**: In the table provided for the effect of training length, 15k training steps correspond to 10 epochs for 3k training images. To facilitate comparison, we report the runtime **per epoch** below. All values are in minutes.
> > >
> > > | Method   | PixArt (3k images) | FLUX (3k images) | SDXL (1.5k images) |
> > > | -------- | ------------------ | ---------------- | ------------------ |
> > > | **Ours** | 4.25| 10.52| 1.40 |
> > > | Self-Dis | N/A| N/A| 1.28|
> > > | H-Guide  | N/A| N/A | 1.51|
> > >
> > > **3.**  We thank the reviewer for this important point. Our primary emphasis is that the learning signal in our method does not rely on specific wording variations in the prompt, but rather on consistent semantic differences under matched conditions. In practice, we observe that the learned concept vectors generalize well across multiple neutral prompts such: "a person", "a portrait of a person", and "a human subject", indicating that the method captures a stable semantic direction rather than overfitting to particular phrasing.
> > >
> > > That said, we agree that our method is best suited to visually grounded attributes that admit a reasonably consistent semantic direction, such as gender, age, style, or motion. For more complex linguistic phenomena such as negation (e.g., "a person who is not smiling") or abstract relations, the assumption of clean semantic decomposition becomes less reliable, and the alignment signal may be weaker or ambiguous. We also note that such cases have not been explicitly investigated in prior baselines, which similarly focus on positively defined, visually grounded attributes. We will explicitly include a limitations section in the revised manuscript to clarify these boundaries, supported by visual examples. In limitations section, we will discuss: the additional training overhead compared to prompt-only methods, reduced effectiveness for prompts involving negation or abstract relations, and conditions under which the observed near-linearity degrades, as also noted in Q5 of reviewer LUbo.
> > >
> > > **4.**  We clarify that Table 2 is not intended to show bias mitigation, but rather to evaluate quality preservation under broader semantic edits. The near-identical CLIP and AS values indicate that our method does not degrade image quality or prompt alignment, while still enabling the intended transformations (as shown qualitatively in Appendix F).

---

### Decision · Program_Chairs · 2026-04-30

**Decision:**

Accept (regular)

**Comment:**

The paper presents a model agnostic framework for discovering interpretable and linearly controllable semantic attributes across any text-to-image diffusion model backbone, targeting fair and safe image generation.

The paper received all Weak Reject scores pre-rebuttal. The rebuttal well-addressed most of reviewers' concerns. Reviewers G4gB and TEJ2 increased their scores to Weak Accept. Reviewer fDDu, while did not change the score, mentioned in the Final Justification that his final decision was Weak Accept. Only Reviewer LUbo kept his score unchanged. The AC checked and found the rebuttal convincingly addressed the questions raised by Reviewer LUbo.

Based on the positive reviews and the contributions of the work, the AC decided to accept the paper.